# BRCA2 deficiency and replication stress drive APOBEC3-Mediated genomic instability

Kathy Situ[1,2,12], Haohui Duan[1,2,12], Stephen K. Godin[1,2], Joshua Yang[1,2], Gabrielle Q. McCloskey[2], Basim Naeem[2], Margaret K. Gillis[2], Muhammad H. Zeb [2], Silvi Salhotra[2], Pratha Rawal[2], Nisha Patel[2], Salome K. Mouliere[2], Jie Chen[3], Angéla Békési [4,5], Hajnalka L. Pálinkás[4,5], Subramanian Venkatesan [6], Abby M. Green [7], Nicolai J. Birkbak [8,9], Beáta G. Vértessy [4,5], Charles Swanton [6,10,11] & Shailja Pathania [1,2] ✉

BRCA2 plays a critical role in stabilizing stalled replication forks, yet critical gaps remain in understanding how BRCA2 deficiency triggers fork collapse and drives genomic instability. Here, we identify cytidine deaminase APOBEC3B as a key driver of this process. Using a unique uracil-in-DNA probe, we show that BRCA2 loss promotes APOBEC3B-mediated uracil accumulation in single-stranded DNA (U-ssDNA) at stalled forks. These lesions when processed by UNG2 and APE1, trigger fork collapse and release ssDNA fragments into the cytoplasm, activating NF-κB signaling. This in turn upregulates APOBEC3B expression, establishing a self-reinforcing loop that amplifies cytidine deamination at stalled forks and exacerbates genomic instability. Depletion of APOBEC3B, UNG2, or APE1 rescues these defects. Notably, BRCA1-deficient cells do not accumulate U-ssDNA or induce APOBEC3B under replication stress, highlighting a BRCA2-specific vulnerability. Clinically, low APE1 expression correlates with poor survival in patients with *BRCA2*-mutant tumors, with high APOBEC3 levels further worsening outcomes. Together, our findings establish that replication stress, whether intrinsic or therapy induced, triggers APOBEC3B overexpression and potentially activates an APOBEC3B-driven mutagenic loop in BRCA2-deficient cells. These results position APOBEC3B, UNG2 and APE1 as critical regulators of *BRCA2*-mutant tumor evolution and therapy resistance.

BRCA2 is a tumor suppressor essential for maintaining genomic stability. Germline mutations in *BRCA2* greatly increase the risk for breast, ovarian, prostate, and pancreatic cancer[1–3]. BRCA2 protects genome integrity mainly through its role in DNA damage repair, including homologous recombination (HR)- driven double-strand break (DSB) repair[4,5], inter-strand crosslink repair[6–8], and stabilization of stalled replication forks[9–14]. Notably, BRCA2 deficiency leads to persistent accumulation of single-stranded DNA (ssDNA) at stalled forks due to defective fork protection mechanisms[9,10,13]. While ssDNA accumulation

is a recognized consequence of BRCA2 loss, it remains unclear whether this intermediate directly contributes to fork collapse and genomic instability observed in BRCA2-deficient cells.

A critical vulnerability of ssDNA is its susceptibility to cytidine deamination[15], especially by members of the APOBEC3 family (Apolipoprotein B mRNA-editing enzyme catalytic polypeptide-like 3)[16]. These enzymes, especially APOBEC3A (A3A) and APOBEC3B (A3B), deaminate cytosines to uracils in DNA and have emerged as major drivers of mutational signatures across human cancers[17–19]. While

uracils in double-stranded DNA (dsDNA) are typically repaired faithfully via the base excision repair (BER) pathway, their presence in ssDNA poses a greater threat. DNA glycosylases like UNG2 remove the uracil, creating abasic sites (AP sites)[20,21], which are acted upon by APE1 endonuclease. In dsDNA, this reaction is typically followed by faithful gap filling via DNA polymerase beta using the opposite strand as a template. However, in ssDNA, APE1-mediated cleavage of AP sites leads to the formation of double-strand DNA breaks, a significant contributor to genomic instability. Though protection of AP sites on ssDNA by proteins such as HMCES[22], the Shu complex, and RAD51[23,24] helps guard them from cleavage, these protective mechanisms may falter under the excessive ssDNA burden caused by BRCA2 deficiency. This raises the possibility that APOBEC3-driven cytidine deamination of the replication stress-induced ssDNA is a major driver of fork collapse and genomic instability in BRCA2-deficient cells.

Mutagenesis driven by APOBEC3A (A3A) and APOBEC3B (A3B) is evident across tumor genomes, implicating their deaminase activity in cancer initiation and progression[17,18,25–32]. Recent studies have shown that induction of A3A and A3B leads to genomic instability, often marking an early step in tumor development[17,33]. Despite the established role of APOBEC3s in cancer evolution, their interplay with BRCA2 loss remains unexplored. Are BRCA2-deficient cells especially susceptible to APOBEC3 activity upon replication stress? Could the convergence of fork instability, persistent ssDNA accumulation, and APOBEC3 induction create a mutational "perfect storm" unique to BRCA2-deficient cells?

In this study, we demonstrate that replication stress in BRCA2-deficient cells triggers a self-amplifying cycle of genomic instability driven by APOBEC3B. We identify APOBEC3B, together with BER enzymes UNG2 and APE1, as key mediators of uracil processing in single-stranded DNA (ssDNA) at stalled forks. Their coordinated activity converts deaminated cytosines into DNA breaks, thereby exacerbating replication fork collapse. Furthermore, we show that ssDNA generated at stalled forks in BRCA2-deficient cells not only serves as a substrate for APOBEC3B-mediated cytidine deamination, but upon fork collapse and release into the cytoplasm, also activates NF-κB signaling to induce further APOBEC3B expression. This creates a damaging feed-forward loop that destabilizes stalled forks and amplifies genome instability. Interestingly, this mechanism is specific to BRCA2 loss and not evident upon BRCA1 deficiency, underscoring distinct biological consequences of mutations in these genes[34–37]. Our findings make the case that APOBEC3B is not merely a passive mutator but a major orchestrator of replication stress-induced genomic instability in BRCA2-mutant cancers.

## Results

### BRCA2 deficiency promotes replication stress-induced uracil accumulation

Stalled replication forks accumulate ssDNA coated with replication protein A (RPA), a critical intermediate during stalled replication fork repair, for recruiting and activating various repair and checkpoint proteins to the stalled forks[38–40]. However, persistent and unresolved accumulation of ssDNA at stalled forks could result in genomic instability[10,41–43]. We have previously shown that BRCA2 loss leads to persistent accumulation of RPA-coated ssDNA at stalled replication forks[10,44]. Given that ssDNA is particularly vulnerable to cytosine deamination, we asked whether replication stress in BRCA2-deficient cells promotes the accumulation of uracil within these ssDNA regions (hereafter referred to as U-ssDNA). Cytosine deamination in ssDNA is especially problematic because uracil in ssDNA, when excised by UNG2, creates an abasic site, which, when cleaved by APE1, could result in a double-strand break, a prime contributor to genomic instability.

To address whether BRCA2-deficient cells are especially susceptible to accumulation of U-ssDNA upon replication stress, we

developed a U-ssDNA detection assay using a catalytically inactive UNG as a probe (3XFLAG-ΔUNG)[45] (Fig. 1a and Supplementary Fig. 1a). This mutant (ΔUNG) binds uracil in DNA (U-DNA) but cannot excise the uracil, thus staying bound to it[45,46]. The U-DNA bound 3XFLAG-ΔUNG can be detected with a FLAG antibody, providing a measure of cellular uracil content. Specificity of the ΔUNG probe to uracil was confirmed using 5-fluorodeoxyuridine (5-FdUR), a drug that interferes with thymidylate biosynthesis and leads to elevated levels of uracil in genomic DNA[45,47]. As shown in Supplementary Fig. 1b, c, 5-FdUR-treated cells showed U-DNA foci with the ΔUNG probe but not the untreated cells, confirming the probe's specificity.

We next asked whether BRCA2 deficiency leads to U-ssDNA accumulation upon replication stress. The standard uracil detection protocol calls for using the ΔUNG probe on denatured DNA to unmask the uracil epitope in DNA[45,46]. To selectively detect uracil in ssDNA, we modified the assay by eliminating the DNA denaturation step (Fig. 1a). Using this modified assay, we observed a significant increase in U-ssDNA accumulation following treatment with cisplatin or hydroxyurea (HU) in cells treated with two independent siRNAs targeting BRCA2 (Fig. 1b–h). The data presented here is for the number of U-ssDNA foci/cell. Similar differences were also observed when measuring the intensity of U-ssDNA foci between these groups (Supplementary Fig. 1d, e). We extended these findings to BRCA2 mutant tumor lines, PEO1 (ovarian) and CAPAN1 (pancreatic) (Supplementary Fig. 1f, g). Both these tumor lines are BRCA2 mutant and have undergone BRCA2 loss of heterozygosity (LOH)[48–50]. Both tumor lines showed high intensity of U-ssDNA staining (foci count was not possible) (Fig. 1i–k). Notably, their isogenic revertant clones (PEOC4 and C2-12, respectively), wherein the BRCA2 expression is restored[48–50] (Supplementary Fig. 1f, g), showed reduced U-ssDNA accumulation upon replication stress (Fig. 1i–k). These findings were further validated in additional cell lines, including the U2OS line with doxycycline (DOX)-inducible BRCA2 shRNA (Fig. 1l–o), and BRCA2-deficient BICR6 cells (Supplementary Fig. 1h–j). These findings collectively provide compelling evidence that BRCA2 deficiency leads to replication stress-induced uracil accumulation.

### Replication stress-induced ssDNA in S-phase is the substrate for uracil accumulation in BRCA2-deficient cells

We next investigated whether the U-ssDNA accumulation observed in BRCA2-deficient cells occurs specifically in response to replication fork stalling in S-phase cells. To mark S-phase cells, we used PCNA, which has been shown to localize to active and stressed/stalled replication forks[51–53]. Co-staining BRCA2-deficient cells with PCNA and ΔUNG U-DNA probe revealed that the majority of cells exhibiting U-ssDNA signal after replication stress induced by HU were also PCNA-positive, indicating that uracil incorporation occurs predominantly in S-phase (Fig. 2a, b). This co-positivity of the PCNA and U-ssDNA signal was more readily detected in HU-treated cells compared to cisplatin-treated cells, likely because cisplatin-treated cells tend to exit S-phase during 24h recovery, albeit with uracil in DNA. In the HU-treated cells, we also observed strong co-localization between U-ssDNA and PCNA foci in around 20–25% cells, suggesting that U-ssDNA is associated with stalled replication forks. Quantification of the number of colocalized PCNA and U-ssDNA foci per cell following HU treatment is shown in Fig. 2c.

BRCA2-deficient cells treated with cisplatin and HU also showed a significant increase in cells co-positive for U-ssDNA and pRPA32 (phosphorylated RPA32), a marker of ssDNA at stalled forks (Supplementary Fig. 2a). Similar results were observed in BRCA2 mutant tumor lines CAPAN1 and PEO1, which showed increased U-ssDNA and pRPA32 co-staining compared to their isogenic BRCA2-revertant counterparts C2-12 and PEOC4, respectively (Supplementary Fig. 2b, c). The strong correlation between an increase in RPA32 signal (reflecting increased ssDNA) and U-ssDNA in BRCA2-deficient cells

 

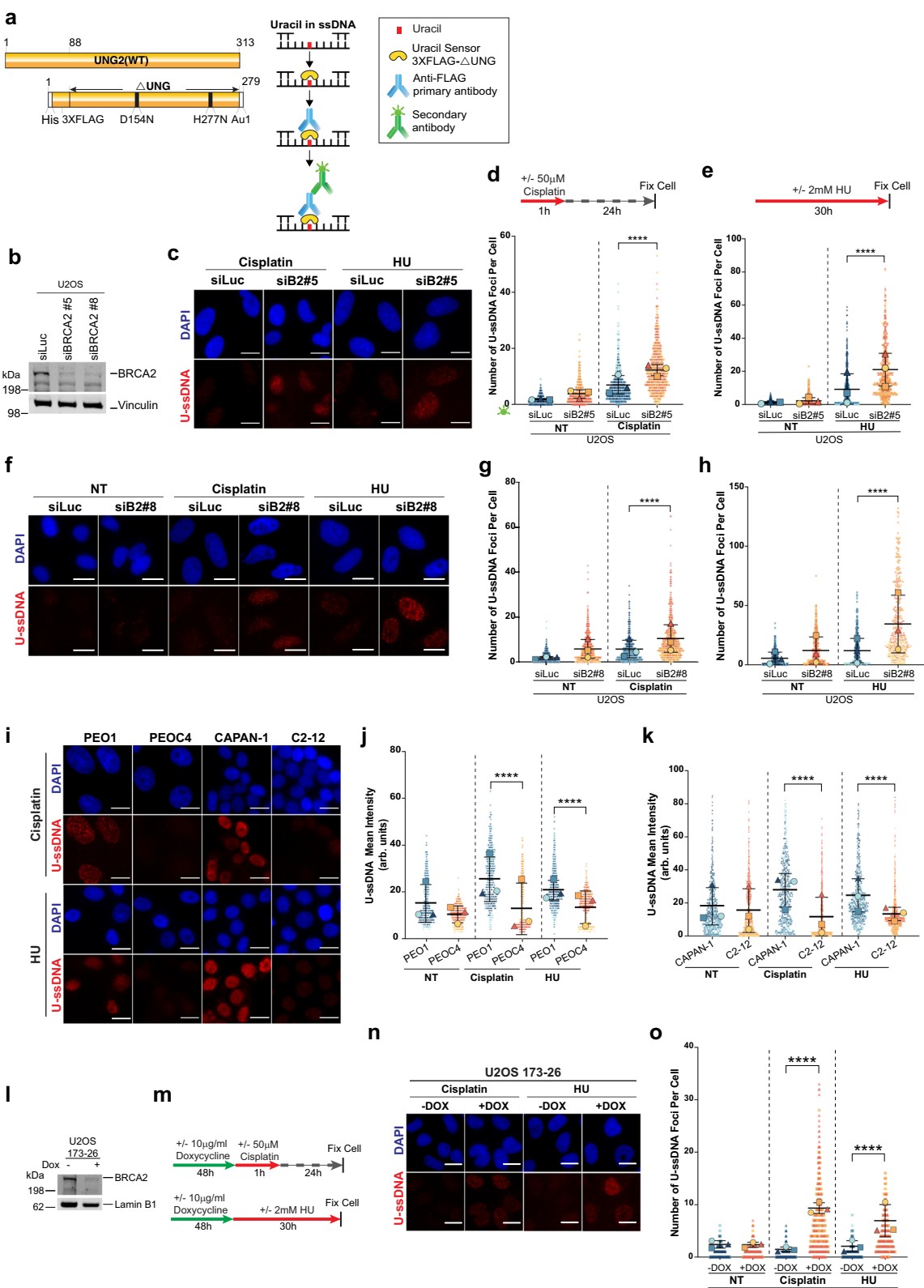

suggests that replication stress-induced ssDNA is a key substrate for uracil accumulation.

To directly test whether uracil accumulation occurs on ssDNA, we treated BRCA2-deficient cells and *BRCA2* mutant tumor cells (PEO1) with S1 nuclease, which degrades ssDNA. S1 nuclease treatment significantly reduced the U-ssDNA signal in cells treated with HU or cisplatin (Fig. 2d and Supplementary Fig. 2d, e), confirming that ssDNA is

the primary substrate for uracil accumulation. Given that S1 nuclease can also act on RNA, we included a control treatment using RNaseA to determine whether RNA contributes to the uracil signal. RNaseA pre-treatment had no effect on the uracil signal, in contrast to the loss of signal observed after S1 nuclease treatment (Supplementary Fig. 2f–h), confirming that uracil accumulation in BRCA2-deficient cells upon replication stress is specific to ssDNA.

**Fig. 1 | Replication stress induces uracil accumulation in BRCA2-deficient cells. a** Schematic of the full-length UNG2 protein, the probe (3XFLAG-ΔUNG), and the method used for U-ssDNA detection. **b** Western blot analysis of BRCA2 in cells transfected with two independent BRCA2 siRNAs (siB2#5 or siB2#8); vinculin serves as a loading control. **c, f** Representative images of U-ssDNA accumulation in U2OS cells transfected with the indicated siRNAs. Cells were treated with cisplatin (50 μM, 1 h; collected 24 h post recovery) or HU (2 mM, 30 h) and stained with ΔUNG probe under nondenaturing conditions. **d, e, g, h** Quantification of U-ssDNA foci/cell for (**c, f**). Foci were analyzed in Image J; 100–200 cells per replicate. SuperPlots[131] of three independent biological replicates (n = 3 are plotted. Highlighted shapes (circle, square, and triangle) represent the average of each replicate, and black lines represent mean ± SD. **i** U-ssDNA accumulation in PEO1/PEOC4 or CAPAN-1/C2-12 cells. Cells were not-treated (NT) or treated with cisplatin and HU as in (**c, f**). **j, k** Quantification of U-ssDNA intensity (arbitrary units) per cell detected with ΔUNG probe in PEO1/PEOC4 (j) or CAPAN-1/C2-12 (**k**) cells; > 200 cells were

analyzed per replicate. SuperPlots are plotted as described for (**d**). **l** Western blot analysis of BRCA2 in U2OS cells containing a doxycycline (DOX)-inducible shRNA against BRCA2. Cells were treated with or without doxycycline (10 μg/mL, 72 h). LaminB1 is the loading control. **m** Schematic of doxycycline, cisplatin, and HU treatment. **n** U-ssDNA foci in U2OS cells with DOX-inducible BRCA2 shRNA. Cells were treated with doxycycline for 72 h prior to cisplatin (50 μM, 1 h; 24 h recovery) or HU (2 mM, 30 h). **o** Quantification of U-ssDNA foci. SuperPlots are plotted as described for (**d**). Statistical significance for all the sets of three independent biological replicates (n = 3) analyzed above was determined using a repeated-measurement model followed by two-tailed multiple comparisons with a Bonferroni post hoc test. ****p ≤ 0.0001; *p ≤ 0.05; ns, not significant. In all images, the scale bar represents 20 μm. Western blot images presented here are representative of three or more western blots with similar results. Source data are provided as a Source Data file.

## BRCA1 deficiency does not induce U-ssDNA accumulation upon replication stress

Both BRCA1 and BRCA2 are key players in stalled replication fork repair, and their loss leads to fork degradation and genomic instability[9,38,54]. However, while BRCA2-deficient cells accumulate U-ssDNA under replication stress, our data reveal that BRCA1-deficient cells do not exhibit this phenotype. Specifically, no increase in U-ssDNA after HU or cisplatin was observed in BRCA1-depleted cells following treatment with HU or cisplatin (Fig. 2e–h). This finding was further confirmed in a *BRCA1* mutant ovarian tumor line (UWB1.289) (Supplementary Fig. 2i) and in cells depleted of BRCA1 with an additional siRNA targeting BRCA1 (Supplementary Fig. 2j–k). In these experiments, BRCA2-deficiency consistently led to elevated U-ssDNA accumulation. Reduced U-ssDNA in BRCA1-deficient cells correlated with fewer RPA foci (a marker of ssDNA) compared to BRCA2-deficient cells following HU or cisplatin treatment (Fig. 2i, j). This is in keeping with previous reports showing that BRCA1-deficient cells accumulate less ssDNA than BRCA2-deficient cells upon HU-induced replication fork stalling[10,38]. Differential U-ssDNA phenotypes observed between BRCA1- and BRCA2-deficient cells likely reflect mechanistic differences in how ssDNA is generated and resolved at stalled forks under replication stress in these cells.

## Abasic site accumulation upon replication stress in BRCA2-deficient cells is UNG2 dependent

Removal of damaged base pairs or DNA lesions by glycosylases is known to generate apurinic/apyrimidinic sites (AP sites), also known as abasic sites[55,56]. Given the higher accumulation of U-ssDNA in BRCA2-deficient cells, we asked whether there is higher AP site accumulation in BRCA2-deficient cells undergoing replication stalling. To quantify AP sites, we used a biotinylated aldehyde reactive probe (ARP) assay that covalently links to AP sites on DNA[57–59]. We used methyl methanesulfonate (MMS), a known inducer of AP sites to confirm the specificity of this assay[60–62]. As expected, cells treated with MMS show a robust increase in ARP signal, signifying an increase in abasic (AP) sites (Supplementary Fig. 3a, b). Methylene blue serves as a DNA loading control in these assays. Treatment of BRCA2-deficient cells (two different BRCA2-targeting siRNAs) with cisplatin or HU resulted in a significant increase in AP sites compared to the WT cells (Fig. 3a–c). Similarly, *BRCA2* mutant tumor lines PEO1 and CAPAN1 also showed an increase in AP sites after cisplatin and HU treatment, compared to their BRCA2 restored counterparts, PEOC4 and C2-12, respectively (Fig. 3d, e). This was also confirmed in DLD1-BRCA2$^{-/-}$ cells (Supplementary Fig. 3c, d). In keeping with low U-ssDNA accumulation in BRCA1-deficient cells, AP sites were also not significantly increased in cisplatin or HU-treated BRCA1-deficient cells (BRCA1 targeted by two different siRNAs) (Fig. 3f, g) or in *BRCA1* mutant tumor line UWB1.289 (Fig. 3h). Together these data confirm that BRCA2-deficient, but not BRCA1-deficient cells, are prone to

accumulating AP sites upon replication stress induced by HU or cisplatin.

We next focused on identifying the major glycosylase responsible for the removal of uracil from ssDNA, resulting in the accumulation of abasic sites in BRCA2-deficient cells. Uracil in DNA is specifically recognized by the uracil-DNA glycosylase superfamily (UDG), which includes UNG (UNG1 and UNG2), SMUG1, TDG, and MBD4[21,63,64]. While UNG1 is almost exclusively present in mitochondria, nuclear UNG2 has been identified as the primary glycosylase responsible for removing uracil from both dsDNA and ssDNA[63]. SMUG1 shares substrate specificity with UNG2, and it has been suggested that SMUG1 could serve as a backup for UNG2 in uracil repair[63,65]. TDG and MBD4 exhibit no activity against uracil in ssDNA[63,66,67]. In light of these observations, we asked whether UNG2 and/or SMUG1 play a role in the generation of AP sites in BRCA2-deficient cells. To address this, we co-depleted UNG2 from BRCA2-deficient cells and assessed the accumulation of AP sites after replication stress. As shown in Fig. 3i, j, a significant reduction in AP site accumulation was observed in BRCA2-deficient cells upon UNG2 co-depletion. To further confirm these findings, we used an HCT116 line stably expressing a UNG2 inhibitor UGI[45]. Here too, UNG2 inhibition led to a significant reduction in accumulation of AP sites in BRCA2-deficient cells (Supplementary Fig. 3e, f). In addition, we established a UNG2 knockout (KO) U2OS cell line (Supplementary Fig. 3g) and, similar to siRNA-mediated UNG2 depletion, UNG2 KO also suppressed AP site accumulation in BRCA2-deficient cells treated with cisplatin and HU (Fig. 3k, l).

We next tested the effect of SMUG1-depletion, both alone and in combination with UNG2 loss, on AP site accumulation in these cells. SMUG1 depletion did not reduce AP site accumulation in BRCA2-deficient cells, nor did it add to the suppression observed in UNG2 KO lines (Supplementary Fig. 3h, i) suggesting that under the conditions used in this study, UNG2 is the primary U-ssDNA glycosylase at play in BRCA2-deficient cells.

## UNG2 and APE1 drive stalled fork degradation and genomic instability in BRCA2-deficient cells under replication stress

BRCA2-deficient cells are highly susceptible to DNA breaks and genomic instability when replication forks stall. Given that uracil accumulation in ssDNA at stalled forks, when processed by UNG2 and APE1, can lead to fork collapse, we next asked whether UNG2 and APE1 are major contributors to fork degradation and genomic instability in BRCA2-deficient cells. To assess their impact on stalled fork stability, we used the fork degradation assay. Briefly, cells were pulsed with 5-iodo-2′-deoxyuridine (IdU), followed by 5-chloro-2′-deoxyuridine (CldU) and then treated with HU. The shortening of the CldU tracts following HU treatment served as a measure of fork degradation. As previously reported[9,10,12,68], loss of BRCA2 resulted in significant shortening of CldU tracts compared to control cells after HU treatment, indicating increased fork degradation. Interestingly, we found

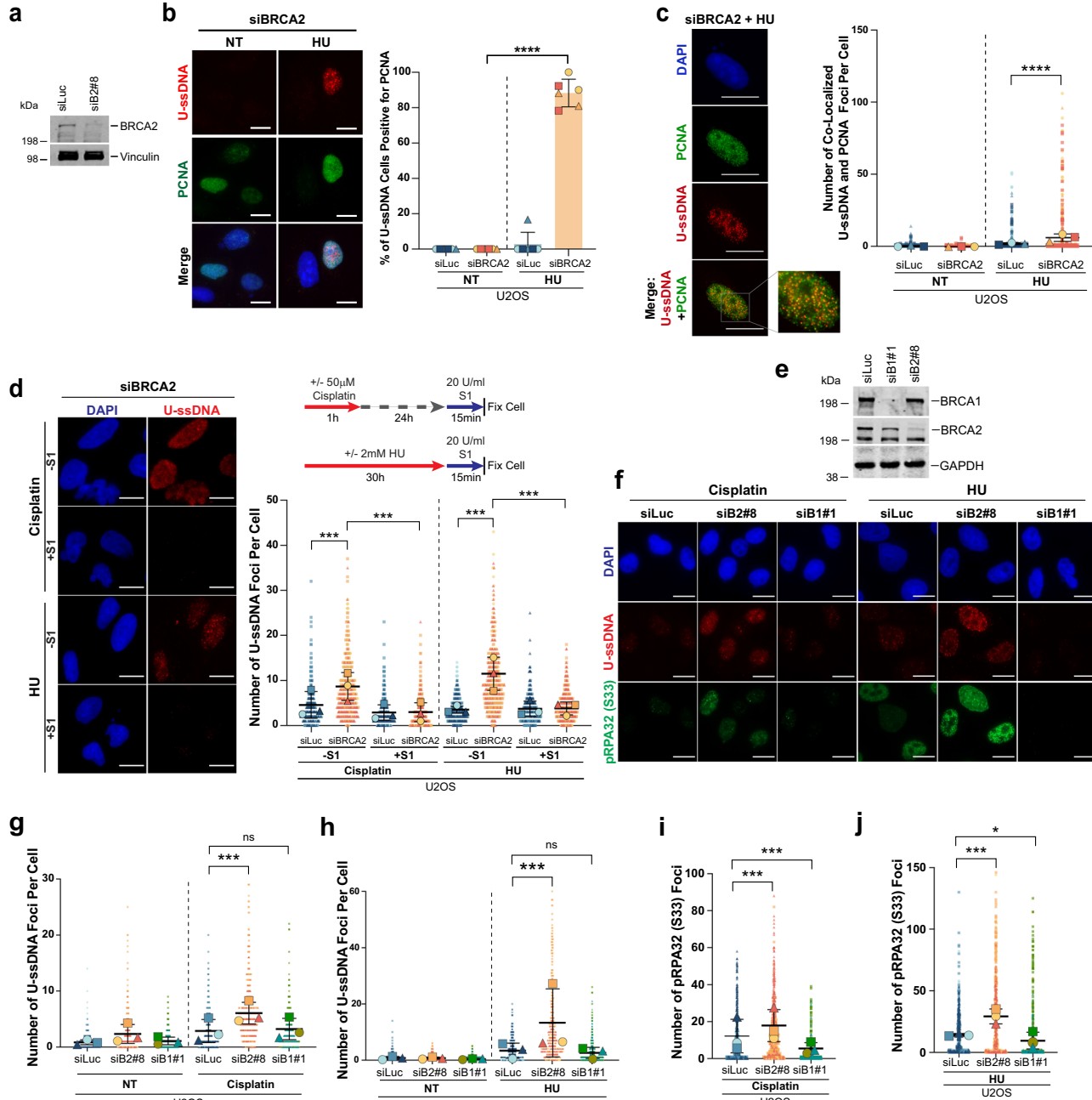

**Fig. 2 | Replication stress-induced ssDNA at stalled forks in BRCA2-, but not BRCA1- deficient cells, is a substrate for uracil accumulation. a** Western blot of BRCA2 in U2OS cells. Vinculin is the loading control. **b** Left: Representative images of U-ssDNA and PCNA immunostaining in BRCA2-deficient cells, untreated (NT) or HU-treated (2 mM, 30 h). Right: Quantification of U-ssDNA-positive cells also positive for PCNA. Cells were transfected with the indicated siRNAs for 48 h before HU treatment. Each highlighted shape (circle, square, and triangle) represents a biological replicate ($n = 3$), each with two technical replicates; >100 cells counted per replicate. Statistical significance was determined by an unpaired two-tailed Student's $t$ test. ****$p \leq 0.0001$. **c** Left: Representative image of U-ssDNA and PCNA foci in a BRCA2-deficient cell treated with HU. Right: Quantification of U-ssDNA and PCNA co-localized foci in U2OS cells with indicated siRNAs. SuperPlots of $n = 3$ biological replicates are shown. The number of co-localized U-ssDNA and PCNA foci per cell (for 100–200 cells) was analyzed through Image J. Each highlighted shape represents the average of each replicate, with the black lines representing the mean ± SD of $n = 3$ independent experiments. Statistical significance was

determined by using a repeated measurement model followed by two-tailed multiple comparisons with a Bonferroni Post hoc test. ****$p \leq 0.0001$. **d** Left: U-ssDNA accumulation in BRCA2-deficient cells treated with or without S1 nuclease. U2OS cells were transfected with indicated siRNAs, treated with cisplatin (50 μM, 1 h, 24 h recovery) or HU (2 mM, 30 h), then incubated with S1 nuclease before fixation and ΔUNG probe staining. SuperPlots are plotted and statistical significance determined as described in (**c**). ***$p \leq 0.001$ (**e**) Western blots for BRCA1 and BRCA2 after transfection with the indicated siRNAs; GAPDH is the loading control.
**f** Representative images of U-ssDNA and pRPA32(S33) foci in U2OS cells after cisplatin or HU treatment. The scale bar represents 20 μm in all the experiments presented above. **g, h** Quantification of U-ssDNA foci for data presented in (**f**).
**i, j** Quantification of pRPA32 (S33) foci for data presented in (**f**). SuperPlots are plotted and statistical significance determined as described in (**c**). ***$p \leq 0.001$; *$p \leq 0.05$; ns: not significant $p > 0.05$. Western blot images presented here are representative of three or more western blots with similar results. Source data are provided as a Source Data file.

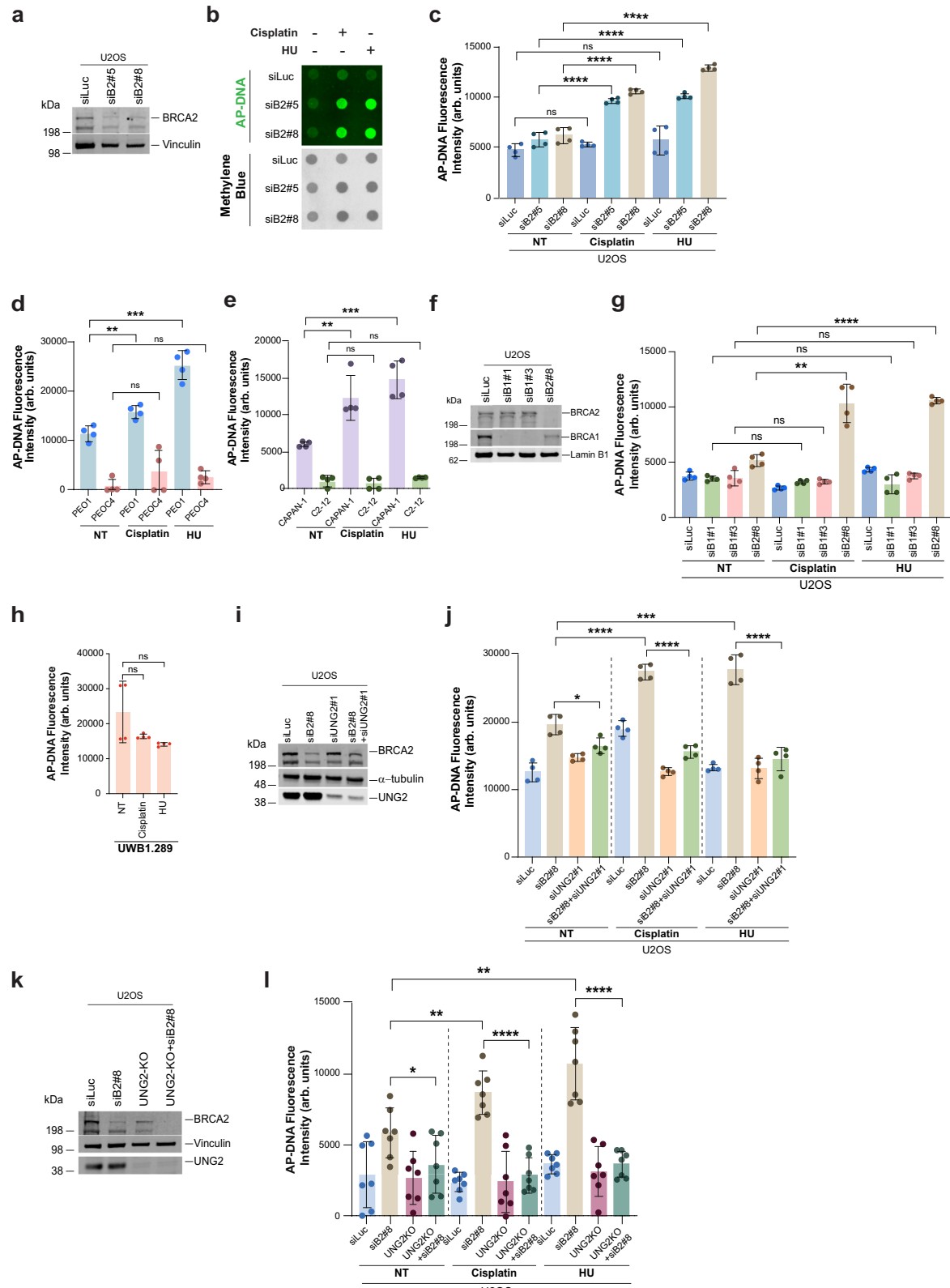

that co-depletion of UNG2 or APE1 resulted in significant rescue of fork degradation in BRCA2-deficient cells (Supplementary Fig. 4a, b, and Supplementary Data 1), indicating that both UNG2 and APE1 contribute to fork degradation in these cells.

To further explore the role of UNG2 and APE1 in fork stability, we used 53BP1 as a marker of DNA damage induced by replication stress[69,70]. Depletion of either UNG2 or APE1 significantly reduced

53BP1 foci in BRCA2-deficient cells treated with cisplatin or HU (Fig. 4a–e and Supplementary Fig. 4c). While 53BP1 can mark diverse types of DNA lesions, prior studies have shown that 53BP1 foci that arise in S- and G2-phase cells after replication stress are distinct from the 53BP1 nuclear bodies formed in G1 cells[71]. Co-staining 53BP1-stained cells with cyclin A (S and G2-phase marker) confirmed that the majority of 53BP1-positive cells in BRCA2-deficient cells are also cyclin

**Fig. 3 | UNG2-driven uracil removal induces abasic site accumulation in BRCA2-deficient cells upon replication stress. a** Western blot of BRCA2 in U2OS cells transfected with two independent BRCA2 siRNAs (siB2#5 or siB2#8); vinculin is the loading control. **b** Dot blot detection of abasic sites (AP-DNA) in U2OS cells transfected with siLuc (control) or siBRCA2 siRNAs and then untreated (NT) or treated with cisplatin (50 μM, 1 h, 24 h recovery) or HU (2 mM, 30 h). Methylene blue served as the loading control. One of four independent experiments is shown. **c** Quantification of AP-DNA intensity normalized to methylene blue. Fluorescence intensity is plotted as arbitrary (arb.) units. Data are mean ± SD, $n = 4$ independent experiments. Statistical significance was determined by an unpaired, two-tailed Student's $t$ test. ****$p ≤ 0.0001$; ns = not significant. **d, e** Quantification of AP-DNA intensity in PEO1/PEOC4 (d) or CAPAN-1/C2-12 (e) cells, treated as in (**b**). Mean ± SD of $n = 4$ independent experiments are plotted. Statistical significance was determined by an unpaired, two-tailed Student's $t$ test. ***$p ≤ 0.001$; **$p ≤ 0.01$; ns = not significant. **f** Western blot of BRCA2 and BRCA1 after siRNA; LaminB1 is a control.

**g** Quantification of AP-DNA intensity in cells transfected with the indicated siRNAs, treated as in (**b**). Data from mean ± SD, $n = 4$ independent experiments. Statistical significance determined by un-paired two-tailed Student's $t$ test. ****$p ≤ 0.0001$; **$p ≤ 0.01$; ns = not significant. **h** Quantification of AP-DNA intensity in *BRCA1* mutant UWB1.289 cells after cisplatin or HU. Data from mean ± SD, $n = 4$ independent experiments. Statistical significance was determined as described for (**g**). **i** Western blot of BRCA2 and UNG2 after siRNA; α-tubulin is the loading control. **j** Quantification of AP-DNA intensity in U2OS cells transfected with indicated siRNAs, treated as in (**b**). Data are mean ± SD of $n = 4$ independent experiments. Statistical significance was determined as described for (**g**). **k** Western blot of BRCA2 and UNG2 in U2OS or UNG2-KO U2OS cells after siRNA. **l** Quantification of AP-DNA intensity after treatment with cisplatin and HU as described in (**b**). Data from mean ± SD of $n = 7$ independent experiments. Statistical significance was determined as described in (**g**). Western blot images presented represent ≥ 3 blots with similar results. Source data are provided as a Source Data file.

A positive (Supplementary Fig. 4d–f), confirming that 53BP1 foci quantified in this study primarily arise in S-phase cells. A role for UNG2 in driving DNA damage in BRCA2-deficient cells was further confirmed in the UNG2 KO U2OS cell line, which showed reduced accumulation of 53BP1 foci following BRCA2 depletion (Fig. 4f–h). Importantly, re-expression of wild-type UNG2 (UNG2- EYFP) restored 53BP1 foci formation in these cells (Fig. 4f–h), directly implicating UNG2 as a major driver of collapsed forks in BRCA2-deficient cells.

To further confirm the contribution of UNG2 and APE1 in driving genomic instability in BRCA2-deficient cells, we assessed micronuclei formation, an established marker of DNA breaks. Depletion of either UNG2 or APE1 significantly suppressed micronuclei formation in BRCA2-deficient cells treated with HU or cisplatin (Fig. 4i–m, western blots in Supplementary Fig. 4g, h). This phenotype was validated with additional siRNAs targeting UNG2 and APE1 (Supplementary Fig. 4i–l), and was confirmed in the UNG2 KO cell line as well. Notably, re-expression of wild-type UNG2 restored micronuclei formation in UNG2KO BRCA2-deficient cells (Fig. 4n, o). Consistent with these findings, depletion of UNG2 and/or APE1 also reduced DNA breaks, as measured by neutral comet assay (Fig. 5a, b). Of note, the addition of exogenous uracil DNA glycosylase (UDG) significantly increased comet tails specifically in BRCA2-deficient cells, confirming the increased presence of uracil in BRCA2-deficient cells, presumably on ssDNA, which, when excised by UDG, results in abasic sites sites which under alkaline conditions are converted to double-strand breaks (Supplementary Fig. 5a–c).

Consistent with the role of UNG2 in driving stalled fork instability, DSBs, and micronuclei formation in BRCA2-deficient cells, its depletion in BRCA2-deficient cells rescued their sensitivity to replication stress-inducing agents (Fig. 5c–e). Similarly, knockout of UNG2 reduced the sensitivity of BRCA2-deficient cells to HU and cisplatin (Fig. 5f–h). Similar rescue was observed upon co-depletion of APE1 (Fig. 5i–k). These findings were validated using additional UNG2- and APE1- targeting siRNAs (Supplementary Fig. 5d–i). Furthermore, stable knockdown of APE1 (shAPE1) in HeLa cells significantly rescued the survival of BRCA2-deficient cells upon replication stress. Notably, re-expression of an siAPE1-resistant APE1 cDNA and shAPE1-resistant APE1 resensitized these cells to cisplatin and HU (Supplementary Fig. 5j–m), further confirming APE1's role in mediating replication stress-induced toxicity in BRCA2-deficient cells.

Collectively, these findings show that loss of UNG2 or APE1 prevents stalled fork degradation, suppresses DSBs, and reduces micronuclei accumulation in BRCA2-deficient cells. This suggests that UNG2 and APE1 levels may influence the response of *BRCA2* mutant tumors to chemotherapy, particularly to platinum-based therapy that induces replication stress. We tested this hypothesis by analyzing a dataset of patients with pathogenic *BRCA2*-mutant ovarian tumors and found that low APE1 expression in the tumors correlated with worse progression-free survival in *BRCA2*-mutation carriers (Fig. 5l). No such correlation was observed for *BRCA1*-mutation carriers (Supplementary Fig. 5n).

## APOBEC3B drives uracil accumulation and genomic instability in BRCA2-deficient cells

APOBEC3B (A3B) is a key enzyme that drives cytosine deamination on ssDNA[72–75]. To determine whether A3B contributes to U-ssDNA accumulation and genomic instability in BRCA2-deficient cells under replication stress, we depleted A3B using both siRNA-mediated knockdown and CRISPR-generated knockout clones (single A3B KO as well as A3A/A3B double knockout (DKO) lines) (Supplementary Fig. 6a, b), enabling us to directly examine the contribution of A3B to U-ssDNA accumulation in the BRCA2-deficient background. Loss of A3B resulted in a near complete suppression of U-ssDNA accumulation in BRCA2-deficient cells under replication stress (Fig. 6a–c and Supplementary Fig. 6c–e, h, i). A3B depletion was confirmed by qPCR and western blot analysis (Supplementary Fig. 6f, g). To further address the role of A3B in U-ssDNA accumulation, we transfected A3A/A3B DKO cells with either wild-type (WT) or catalytically inactive A3B (A3B-E255Q) (Supplementary Fig. 6j). As shown in Fig. 6d, e, transfection with WT A3B, but not the catalytically inactive mutant (E255Q), significantly restored U-ssDNA formation in BRCA2-deficient cells upon replication stress. This strongly implicates the deaminase activity of A3B in driving U-ssDNA accumulation in BRCA2-deficient cells.

We next tested whether inhibiting U-ssDNA formation through depletion of A3B would block fork degradation and DNA breaks in BRCA2-deficient cells. Indeed, loss of A3B significantly blocked fork degradation (Supplementary Fig. 6k, l and Supplementary Data 2) in BRCA2-deficient cells. In keeping with no accumulation of U-ssDNA in BRCA1-deficient cells, A3B co-depletion in these cells does not rescue stalled fork degradation (Supplementary Fig. 6k, l and Supplementary Data 2). A3B co-depletion in BRCA2-deficient cells also leads to reduced DSBs in the comet assay (Supplementary Fig. 6m) and suppressed micronuclei formation (Fig. 6f, g and Supplementary Fig. 6n). Re-expression of WT A3B in these cells restored micronuclei formation. Here too, expression of the catalytically inactive mutant A3B (E255Q)[76] did not increase micronuclei formation after treatment with cisplatin or HU (Fig. 6h, i). Interestingly, overexpression of WT A3A and not its catalytically inactive version A3A (C106S)[77] in A3A/A3B DKO U2OS cells (U2OS primarily expresses A3B) also induced micronuclei formation, highlighting potential functional redundancy between A3A and A3B (Fig. 6h, i). Consistent with its role in driving genomic instability, A3B depletion rescued the cell sensitivity of BRCA2-deficient cells to HU and cisplatin (Fig. 6j–l). However, in the *BRCA2*-mutant tumor line PEO1, A3B loss did not significantly rescue drug sensitivity (Supplementary Fig. 6o–q). We speculate that chronic APOBEC3 activity in long-established BRCA2 tumor lines like PEO1 may have led to compensatory changes or selection of downstream effectors that can help these cells bypass APOBEC3-induced DNA damage. In addition, the inherently high baseline genomic instability in *BRCA2*-mutant tumor lines may render them intrinsically hypersensitive to HU and cisplatin, limiting the extent of rescue achievable by siRNA-mediated A3B depletion alone.

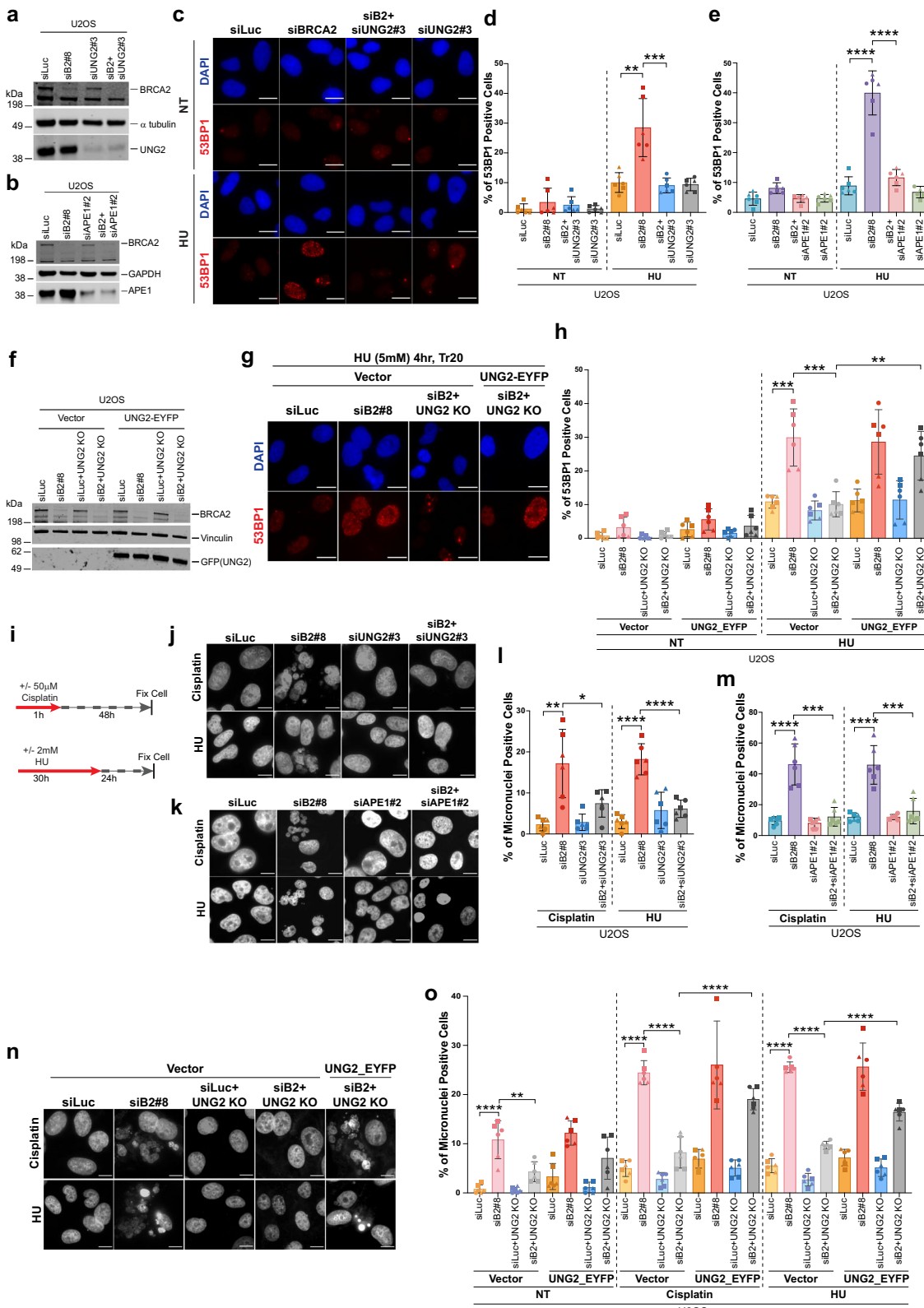

Nevertheless, the consistent and significant rescue of genomic instability and drug sensitivity in BRCA2-depleted cells upon A3B knockdown strongly supports the conclusion that A3B, and A3A when expressed, act as major drivers of U-ssDNA accumulation, fork degradation, and genomic instability in BRCA2-deficient cells under replication stress.

## BRCA2 deficiency induces APOBEC3B overexpression upon replication stress and ssDNA accumulation

We next investigated the relationship between replication stress and A3B expression in BRCA2-deficient cells. Surprisingly, we observed a significant upregulation of A3B at both mRNA and protein levels in BRCA2-deficient cells upon replication stress induced by HU or

**Fig. 4 | UNG2 and APE1 drive genomic instability in BRCA2-deficient cells.**
**a**, **b** Western blots of BRCA2 with UNG2 (**a**) or APE1 (**b**) in U2OS cells transfected with indicated siRNAs; α-tubulin (**a**) or GAPDH (**b**) is the loading control.
**c** Representative images of 53BP1 immunostaining in U2OS cells transfected with the indicated siRNAs, treated with HU (5 mM, 4 h), followed by 18 h recovery. Scale bar represents 20 μm. **d**, **e** Quantification of 53BP1-positive cells showing S-phase specific foci pattern (see Supplementary Fig. 4e). Data are mean ± SD of $n = 3$ independent experiments, each with two technical replicates (> 200 cells/replicate). **f** Western blot of BRCA2 and GFP (UNG2-EYFP) in U2OS or UNG2-KO U2OS cells transfected with the indicated siRNAs. Samples collected 72 h post siRNA transfection. Vinculin is the loading control. **g** Representative images of 53BP1 immunostaining in cells treated as in (**f**). Scale bar represents 20 μm.
**h** Quantification of 53BP1 positive cells with S-phase specific foci pattern. Data are mean ± SD of $n = 3$ independent experiments, each with two technical replicates (> 200 cells/replicate). **i** Treatment schematic for cisplatin or HU treatment in U2OS cells analyzed for micronuclei formation. **j**, **k** Representative images of micronuclei in U2OS cells transfected with the indicated siRNAs and treated with cisplatin and HU. Scale bar represents 20 μm. **l**, **m** Quantification of micronuclei positive cells. Data are mean ± SD of $n = 3$ independent experiments with two technical replicates (150–200 cells/replicate). **n** Representative images of micronuclei in U2OS or UNG2-KO U2OS cells transfected with the indicated siRNAs, followed by transfection with vector or UNG2-EYFP. Cells were treated with cisplatin and HU as in (**i**). Scale bar represents 20 μm. **o** Quantification of micronuclei positive cells for (**n**). Data are shown as mean ± SD of $n = 3$ independent experiments with two technical replicates each. For all the graphs presented here, statistical significance was determined by an un-paired two-tailed Student's $t$ test. ****$p \leq 0.0001$; ***$p \leq 0.001$; **$p \leq 0.01$; *$p \leq 0.05$; ns: not significant $p > 0.05$. Western blot images presented here are representative of three or more western blots with similar results. Source data are provided as a Source Data file.

cisplatin treatment. We see this increase in both U2OS (Fig. 7a, b) and BICR6 (Fig. 7c, d) cell line. We also confirmed an increase in A3B accumulation by immunofluorescence analysis (Fig. 7e). This upregulation in A3B expression upon replication stress was specific to agents like HU and cisplatin, which cause extensive ssDNA accumulation, either through uncoupling of helicase and polymerase or through resection of reversed forks. Agents like olaparib that primarily induce ssDNA gaps[78] in BRCA1 and BRCA2-deficient cells did not induce a similar A3B upregulation in BRCA2-deficient cells (Supplementary Fig. 7a). In addition, A3B upregulation was specific to BRCA2-deficiency, as BRCA1-deficient cells did not exhibit a similar increase (Supplementary Fig. 7b, c). This is in keeping with our observation of reduced U-ssDNA and AP site accumulation in BRCA1-deficient cells compared to BRCA2-deficient cells.

Given that both HU and cisplatin induce ssDNA accumulation in BRCA2-deficient cells, we wondered whether ssDNA from collapsed forks, upon entering the cytoplasm, could serve as a trigger for A3B overexpression. While ssDNA has not been previously linked to A3B induction, it has shown that A3B is upregulated in response to replication stress and DNA damage-inducing agents[26]. In these studies, A3B upregulation is shown to be dependent on activation of the ATR/CHK1 pathway, along with a role for DNA-PKcs[26]. We asked whether extensive accumulation of ssDNA in BRCA2-deficient cells under replication stress contributes to A3B overexpression. If true, introducing exogenous ssDNA into cells should similarly drive A3B overexpression.

To test this, we designed a 65-nucleotide single-stranded oligonucleotide (65nt ssDNA) tagged with a fluorescent (FAM) label. In tandem, we generated a double-stranded DNA (dsDNA) by annealing its complementary strand (65nt dsDNA) (Fig. 7f). We used FACS to confirm similar transfection efficiencies (89–92%) of both ssDNA and dsDNA FAM tagged DNA in our cells (Fig. 7g). Interestingly, ssDNA, but not dsDNA led to a marked increase in A3B expression (Fig. 7h, i). Immunofluorescence analysis further confirmed these findings (Fig. 7j). Together, these results strongly suggest that the accumulation of ssDNA in BRCA2-deficient cells under replication stress drives the upregulation of A3B.

## BRCA2 deficiency induces APOBEC3B-, UNG2-, and APE1-dependent accumulation of cytoplasmic ssDNA

Having established that ssDNA can drive A3B upregulation, we next tested our hypothesis that the extensive ssDNA generated at stalled forks in BRCA2-deficient cells may, upon fork collapse, give rise to ssDNA fragments that are released into the cytoplasm. Once in the cytoplasm, these ssDNA fragments could activate pathways such as NF-κB, thereby inducing A3B expression. To study the accumulation of endogenous cytoplasmic ssDNA, we used an ssDNA-specific antibody in combination with an actin antibody (cytoplasmic marker) (Supplementary Fig. 8a, b). To specifically study ssDNA within the cytoplasm, we used a proximity ligation assay (PLA) with these two antibodies. As

shown in Fig. 8a–c, BRCA2-deficient cells, but not control cells, show a marked increase in PLA puncta in the cytoplasmic region following HU or cisplatin treatment, indicating accumulation of ssDNA in the cytoplasm. No PLA signal was detected when either primary antibody was omitted, confirming the specificity of the assay (Supplementary Fig. 8c). Furthermore, pre-incubation with S1 nuclease eliminated the PLA signal, whereas RNaseA treatment had no effect, confirming that the PLA signal is specific to ssDNA and not RNA (Supplementary Fig. 8d–f). Importantly, co-depletion of A3B, UNG2, or APE1 in BRCA2-deficient cells led to near-complete suppression of cytoplasmic PLA puncta, i.e., accumulation of cytoplasmic ssDNA after HU or cisplatin treatment (Fig. 8d–f). These findings support a model in which APOBEC3B-mediated cytidine deamination at ssDNA regions of stalled forks, followed by UNG2- and APE1-dependent processing, promotes replication fork collapse and the subsequent accumulation of ssDNA fragments into the cytoplasm.

## UNG2 and APE1 activity drive APOBEC3B upregulation in BRCA2-deficient cells upon replication stress

Having established that UNG2 and APE1 drive accumulation of cytoplasmic ssDNA fragments in BRCA2-deficient cells upon replication stress, we next asked whether these enzymes also contribute directly to the upregulation of A3B in BRCA2-deficient cells. This is especially relevant given our finding that ssDNA accumulation promotes A3B expression.

Indeed, co-depletion of either UNG2 or APE1 significantly reduced the replication stress-induced upregulation of A3B in BRCA2-deficient cells (Fig. 8g, h). Consistent with reduced A3B levels, we also observed a significant reduction in U-ssDNA accumulation in UNG2-deficient (UNG2KO) BRCA2-deficient cells following cisplatin or HU treatment (Fig. 8i, j). These findings support a model wherein basal A3B activity in BRCA2-deficient cells initiates limited uracil incorporation into ssDNA at stalled replication forks. Processing of these uracil lesions by UNG2 and APE1 promotes fork collapse and the release of ssDNA fragments into the cytoplasm, where they activate NF-κB signaling to drive further APOBEC3B expression. This creates an amplification loop that not only increases APOBEC3B levels but also leads to heightened U-ssDNA accumulation, thereby reinforcing replication-associated genomic instability. Consistently, BRCA2-deficient cells lacking UNG2 show both reduced APOBEC3B upregulation and significantly diminished U-ssDNA accumulation upon replication stress, supporting a critical role for UNG2 in sustaining this mutagenic feedback circuit.

## NF-κB activation drives APOBEC3B upregulation in BRCA2-deficient cells upon replication stress

APOBEC3 proteins are induced by a variety of triggers, including inflammation, immune signaling and genotoxic stress[30]. Specifically, A3B expression has been linked to activation of the non-canonical NF-

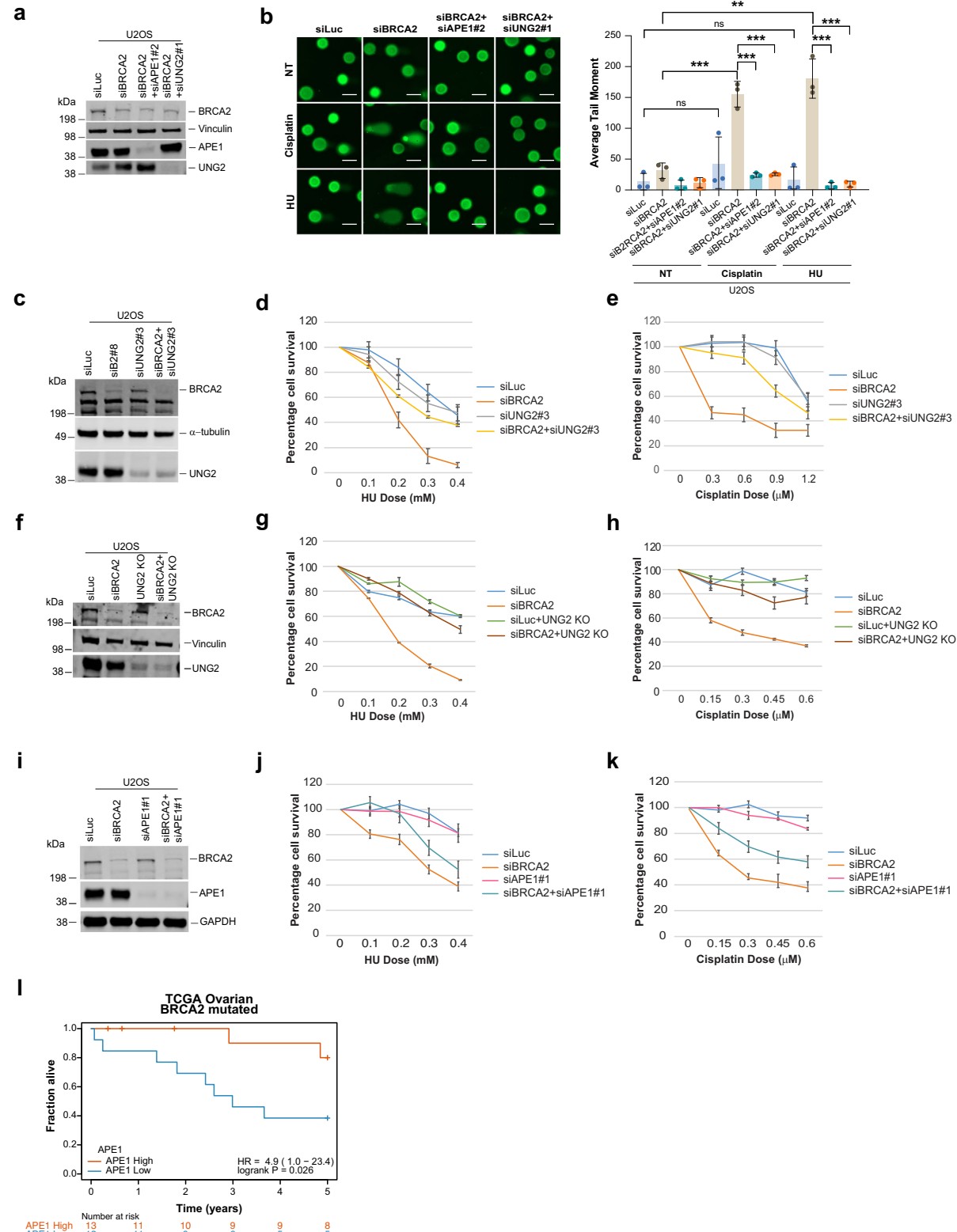

κB pathway[79–81], wherein the translocation of RELB to the nucleus promotes A3B expression[81,82].

To address the role of the NF-κB pathway in A3B upregulation in BRCA2-deficient cells, we monitored the nuclear localization of RELB following replication stress. Upon treatment with cisplatin or HU, BRCA2-deficient cells, but not the control cells (siLuc), showed a significant increase in accumulation of RELB in the nucleus (Fig. 9a–c),

consistent with NF-κB pathway activation. Similarly, ssDNA transfection, but not double-stranded DNA, induced RELB nuclear localization (Supplementary Fig. 9a, b), supporting a model where cytoplasmic ssDNA fragments serve as upstream triggers to upregulate A3B expression. To test whether RELB translocation to the nucleus correlates with A3B induction, we co-stained cells with RELB and A3B-specific antibodies. As shown in Fig. 9d–h (and Supplementary Data 3),

**Fig. 5 | UNG2 and APE1 drive genomic instability in BRCA2-deficient cells.**
**a** Western blot of BRCA2, APE1, and UNG2 in U2OS cells transfected with indicated siRNAs; vinculin is the loading control. **b** Left: Representative images of neutral comet in U2OS cells transfected with the indicated siRNAs and treated with cisplatin or HU. Scale bar represents 50 μm. Right: Quantification of average tail moment. For each experiment, 100–150 comets were analyzed by ImageJ/Open-Comet. Data are mean ± SD of $n = 3$ independent experiments. Statistical significance was determined by an un-paired two-tailed Student's $t$ test. ***$p \leq 0.001$; **$p \leq 0.01$; ns, not significant. **c, f** Western blots of BRCA2 and UNG2 in U2OS (**c**) or UNG2-KO U2OS (**f**) cells transfected with the indicated siRNAs; α-tubulin and vinculin are the loading control. **d, e, g, h** CellTiter-Glo survival assays of U2OS or U2OS

UNG2 KO cells transfected with the indicated siRNAs and treated with HU or cisplatin. Error bars represent SD between triplicates. **i** Western blot of BRCA2 and APE1 in U2OS cells transfected with indicated siRNAs; GAPDH is the loading control. **j, k** CellTiter-Glo survival assay with U2OS cells transfected with the indicated siRNAs and treated with HU (**j**) or cisplatin (**k**). Error bars represent SD between triplicates. **l** Kaplan-Meier curve of 5-year overall survival for ovarian patients with BRCA2 mutant tumors (TCGA dataset). The patients are stratified by median APE1 expression. Mutations associated with the patient data are listed in the Source Data file. Western blot images presented here are representative of three or more western blots with similar results. Source data are provided as a Source Data file.

BRCA2-deficient cells exposed to replication stress show strong co-expression of nuclear RELB and A3B, suggesting that RELB activation correlates with A3B upregulation in BRCA2-deficient cells.

To further confirm the role of the NF-κB pathway in A3B induction, we asked whether inhibiting the NF-κB pathway suppresses A3B expression in BRCA2-deficient cells and suppresses the accumulation of U-ssDNA in these cells. In order to shut down the NF-κB pathway, we used a known NF-κB inhibitor (NF-κBi), BMS-345541. This inhibitor has been shown to target the IKK alpha complex (non-canonical NF-κB)[83]. We used a dose that can selectively inhibit the non-canonical NF-κB pathway and found that incubation with NF-κBi almost completely inhibited A3B expression in BRCA2-deficient cells upon HU treatment (Fig. 9i, j and Supplementary Fig. 9c, d). Of note, NF-κB has been shown to regulate BRCA2 expression[84], which could in part explain the reduction in BRCA2 protein upon NF-κBi treatment in the western blot (Fig. 9j and Supplementary Fig. 9d). This downregulation of BRCA2 protein does not affect the interpretation of this data. Finally, in keeping with reduced A3B expression, NF-κBi also reduces U-ssDNA accumulation in BRCA2-deficient cells (Fig. 9k, l). Together, these findings indicate that the non-canonical NF-κB pathway regulates A3B overexpression in BRCA2-deficient cells upon replication stress.

## APOBEC3B expression influences patient survival in *BRCA2*-mutant ovarian cancer

Our findings suggest that A3B levels may influence how *BRCA2* mutant tumors respond to therapy. While we recognize that APOBEC levels can fluctuate during tumorigenesis and may not precisely reflect their activity in the tumor cells[85], our study does reveal a strong correlation between replication stress and A3B upregulation in BRCA2-deficient cells. This is especially relevant in therapeutic settings, such as platinum-based therapies commonly used for *BRCA2*-mutant cancer, which induce replication stress. Given our observation that cisplatin-induced ssDNA drives A3B expression, we reasoned that this could exacerbate genomic instability in tumors, especially those with high APE1 expression. This prompted us to investigate the impact of APOBEC3 levels in *BRCA2*-mutant tumors stratified by APE1 expression. Interestingly, our analysis of the *BRCA2*-mutant ovarian tumor dataset revealed that tumors with high APOBEC3 levels (combined A3A and A3B) and low APE1 expression were associated with the worst overall survival compared to all other combinations (Fig. 9m).

Together, these observations suggest that APOBEC-driven mutagenesis, particularly in the setting of low APE1 levels, may enable *BRCA2*-mutant tumors to evolve more effectively and acquire mutagenic changes necessary for therapy resistance. This hypothesis is supported by other studies that have described a role for APOBEC in tumor evolution and therapy resistance[29,86–88].

## Discussion

Genomic instability in BRCA2-deficient cells is attributed to defective homologous recombination and defective stalled replication fork repair leading to fork degradation/resection by nucleases such as MRE11 and EXO1[9,12,89]. However, the molecular mechanisms that drive persistently stalled forks to collapse and the downstream

consequences of this collapse are not fully understood. Here, we uncover a previously unrecognized mutagenic cascade in BRCA2-deficient cells, induced by replication stress and driven by APOBEC3B (A3B), UNG2, and APE1. This cascade not only exacerbates fork degradation but also promotes further A3B upregulation, thereby creating a self-amplifying feedback loop that fuels genomic instability in BRCA2-deficient cells.

We show that replication stress in BRCA2-deficient cells leads to excessive accumulation of ssDNA at stalled forks, an ideal substrate for A3B-driven cytidine deamination. While RPA rapidly coats ssDNA at stalled forks, previous studies have shown that APOBEC enzymes can still access and deaminate RPA-coated ssDNA, albeit with reduced efficiency[90–92]. Our findings suggest that replication fork uncoupling or resection at stalled replication forks in BRCA2-deficient cells creates sufficiently large ssDNA regions to enable A3B engagement. The resulting uracil-in-ssDNA (U-ssDNA) is then processed by UNG2 and APE1, resulting in DNA cleavage and fork collapse. APE1 has recently been implicated in fork collapse after A3B activity[93], and in ALC1-deficient cells undergoing replication stress[94]. We go on to show that UNG2- and APE1-mediated processing of U-ssDNA generates ssDNA fragments, which, once in the cytoplasm, activate NF-κB signaling, leading to A3B upregulation. Thus, A3B not only initiates fork collapse but also promotes its own expression via a cytoplasmic DNA–NF-κB axis, amplifying replication-associated genome damage.

This positive self-reinforcement loop (Fig. 10) is especially toxic in BRCA2-deficient cells, which are inherently prone to accumulating excessive ssDNA at stalled forks upon replication stress. While proteins such as HMCES and RAD51 paralogs have been shown to shield AP sites at stalled forks[22–24,95], their protective capacity in BRCA2-deficient setting is likely compromised or overwhelmed. Our data suggest that the accumulation of uracil and AP sites in ssDNA represents a critical vulnerability in these cells.

We identify UNG2 as the primary glycosylase driving uracil removal and AP site accumulation in BRCA2-deficient cells, aligning with its known role in genome-wide uracil repair[96,97]. Although SMUG1 has been reported to process uracil in ssDNA gaps in BRCA2-deficient extracts[24], it did not contribute to genomic instability in our system after HU or cisplatin treatment – both of which primarily cause fork uncoupling and/or fork reversal[13,89,98]. It is possible that the type of uracil containing DNA substrate (dsDNA or extended ssDNA) or its context within replication structures determines glycosylase engagement.

Multiple sources contribute to ssDNA formation at stalled forks, including fork uncoupling, reversed fork resection, and short ssDNA gaps because of PRIMPOL-mediated repriming[41,99–101] or due to defects in Okazaki fragment processing[78]. In BRCA2-deficient cells, fork stalling agents like cisplatin and HU generate long ssDNA regions, which we show are particularly vulnerable to A3B activity. We find that BRCA1-deficient cells do not show A3B induction under similar stress, highlighting a unique vulnerability associated with BRCA2 loss. This likely reflects distinct roles of BRCA1 and BRCA2 in regulating replication fork protection and processing of repair intermediates and warrants further mechanistic investigation. While both BRCA1 and BRCA2-

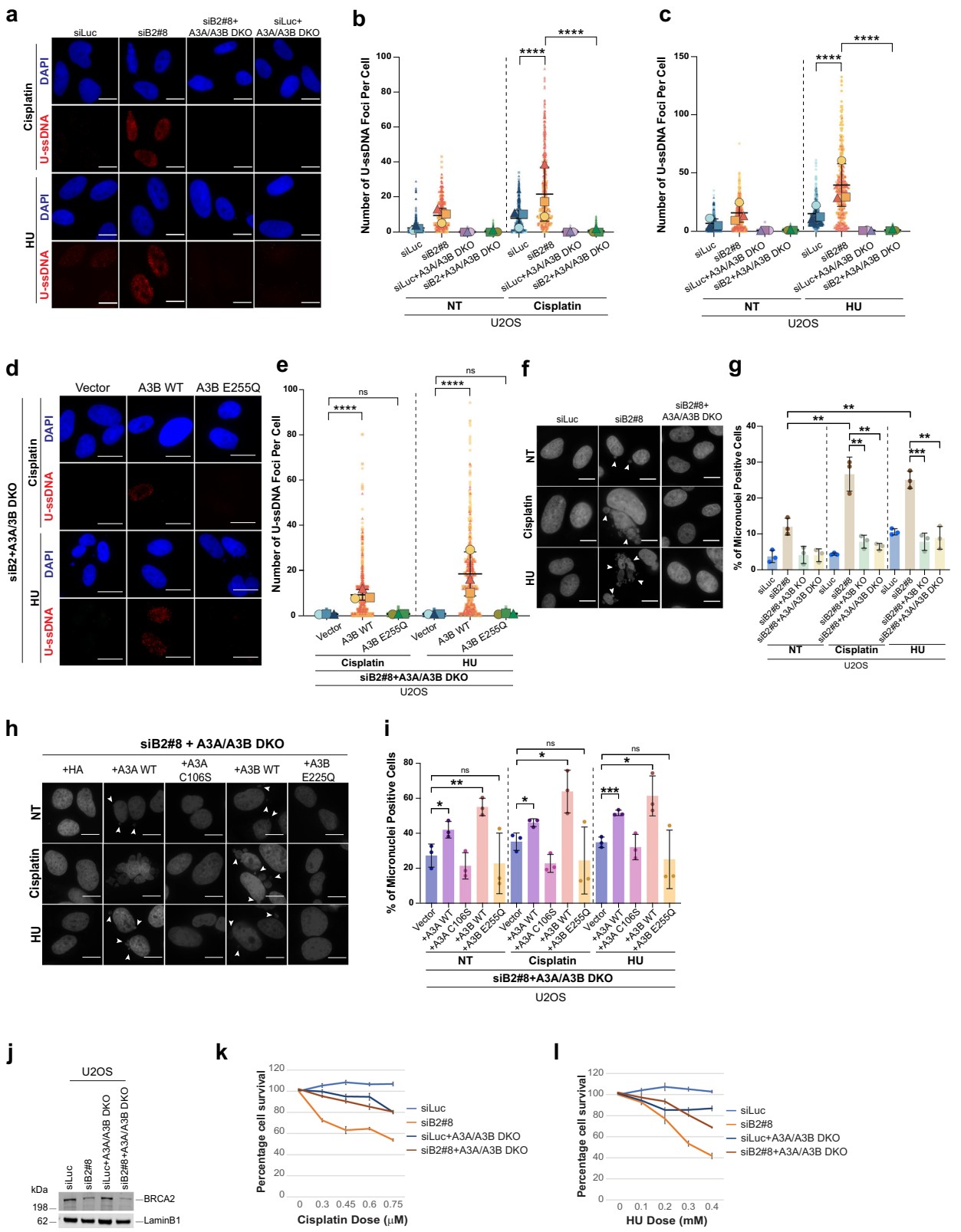

deficient cells can generate internal ssDNA gaps[78,100,102,103], our data indicate that short ssDNA gaps (e.g., from PARPi treatment) may not efficiently trigger A3B-mediated deamination. Instead, long stretches of ssDNA generated by fork uncoupling or reversed fork resection under replication stress may represent the key substrate that initiates this mutagenic loop in BRCA2-deficient cells. Thus, although both BRCA1 and BRCA2 deficiencies lead to genomic instability, our findings

point to an amplified role for A3B in the BRCA2-deficient setting and suggest that BRCA2 loss creates a unique replication stress landscape conducive to APOBEC3B engagement and mutagenic amplification, distinct from BRCA1-deficient settings.

Our findings also provide mechanistic insight into A3B upregulation. While A3B upregulation has been previously linked to viral infection[81,104,105], and genotoxic stress[104], we now show that BRCA2

**Fig. 6 | APOBEC-induced uracil accumulation drives genomic instability in BRCA2-deficient cells undergoing replication stress. a** Representative images of U-ssDNA in wildtype and A3A/A3B DKO U2OS cells transfected with indicated siRNAs. Cells were untreated (NT) or treated with cisplatin (50 μM, 1 h, 24 h recovery) or HU (2 mM, 30 h). **b, c** Quantification of U-ssDNA foci in A3A/A3B DKO U2OS cells transfected with control (siLuc) or BRCA2 siRNA and treated with cisplatin (**b**) or HU (**c**). 100–200 cells were analyzed/replicate. SuperPlots of $n = 3$ independent experiments are presented. Each highlighted shape represents the average of each replicate, with the black lines representing the mean ± SD of the experiments. Statistical significance was determined by a repeated measurement model followed by two-tailed multiple comparisons with a Bonferroni Post hoc test. ****$p \leq 0.0001$. **d** Representative U-ssDNA images of A3A/A3B DKO U2OS cells transfected with the indicated siRNAs, followed by vector, A3B-HA, or A3B E255Q-HA transfection. Treatment with cisplatin or HU was performed 72 h post-siRNA. **e** Quantification of U-ssDNA foci is as in (**b**). ****$p \leq 0.0001$; ns = not significant. **f** Representative micronuclei images in A3B KO or A3A/A3B DKO U2OS cells

transfected with the indicated siRNAs and treated with cisplatin or HU. White arrows mark micronuclei. One of three experiments is shown. **g** Percentage of micronuclei-positive cells in (**f**). Data represent mean ± SD of $n = 3$ independent experiments; > 200 cells/replicate. Statistical significance was determined by unpaired two-tailed Student's t test. ***$p \leq 0.001$; **$p \leq 0.01$. **h** Representative images and (**i**) quantification of micronuclei positive cells in A3A/A3B DKO U2OS cells transfected with the indicated siRNAs and then with vector, A3B-HA, or A3B E255Q-HA, followed with treatment with cisplatin or HU. Micronuclei are indicated by the white arrows. Quantification and statistical significance determined as described in (**g**). ***$p \leq 0.001$; **$p \leq 0.01$; *$p \leq 0.05$; ns = not significant. Scale bars for all images presented here represent 20 μm. **j** Western blot of BRCA2 in U2OS or A3A/A3B DKO U2OS cells transfected with indicated siRNAs. LaminB1 is the loading control. **k, l** CellTiter-Glo survival assay for wildtype or A3A/A3B DKO cells transfected with indicated siRNAs and treated with cisplatin (**k**) or HU (**l**). Error bars represent SD between triplicates. Western blot images represent ≥ 3 western blots with similar results. Source data are provided as a Source Data file.

deficiency uniquely triggers A3B upregulation through cytoplasmic DNA signaling stemming from collapsed forks. Consistent with this, depletion of UNG2 or APE1 in BRCA2-deficient cells reduces fork collapse, cytoplasmic ssDNA accumulation, and NF-κB-dependent A3B upregulation, underscoring the central role for this pathway in A3B-mediated genomic instability. This establishes a tightly coupled self-reinforcing loop that creates a cycle of mutagenesis and genome instability that potentially drives tumor initiation and shapes the mutational landscape of *BRCA2*-mutant tumors.

APOBECs, especially A3A and A3B, are major mutagenic drivers in cancer, with APOBEC signatures detected in around 70% of tumors[16–18,26,29,106]. APOBEC activity has been implicated in the progression of breast, head and neck, lung, prostate, bladder, ovarian, and renal cancer[33,72,107,108], with A3B upregulation observed early in lung and breast tumorigenesis, suggesting a role in cancer initiation[17,26,33]. Although APOBEC3 enzymes are known for driving point mutations (SBS2 and SBS13 signatures)[18,27,28], our data suggest an additional mutagenic outcome in BRCA2-deficient cells. Rather than only leaving a point mutation (SBS2/SBS13) specific mutational footprint, A3B activity in BRCA2-deficient cells may preferentially promote DSBs, thereby driving mutational signatures more consistent with homologous recombination deficiency (HRD).

Clinically, this mutagenic loop initiated by A3B, UNG2, and APE1 in the BRCA2 mutant setting has clear therapeutic implications. *BRCA2* mutant cancer is currently treated with platinum and PARP inhibitors. While these treatments show initial positive responses, resistance inevitably develops[109–112]. One of the factors driving drug resistance is the restoration of stalled fork stability[89,113]. We propose UNG2 and APE1 as modulators of drug sensitivity in BRCA2-deficient cells. Knockdown of UNG2 or APE1 significantly restored stalled fork stability, reduced micronuclei formation, and rescued cell sensitivity of BRCA2-deficient cells to HU and cisplatin. The clinical significance of this pathway is underscored by our analysis of *BRCA2*-mutant tumors, which revealed that low APE1 expression correlates with poor patient survival, and this effect is exacerbated by high APOBEC3 expression. No such correlation was found for *BRCA1*-mutant tumors. These findings suggest that patients with combined APE1-low/APOBEC3-high tumors may represent a high-risk subgroup and could benefit from targeted therapeutic strategies that disrupt this mutagenic feedback loop. Moreover, as replication stress is a common consequence of chemotherapy (e.g., cisplatin) and PARP inhibition, our data raise the possibility that such treatments may inadvertently activate A3B-mediated mutagenesis in BRCA2-deficient tumors, potentially contributing to therapy resistance or tumor evolution. While high APOBEC expression has been associated with both favorable[114–116] and poor outcomes[117–120] depending on the cancer type[121–124], our study underscores the importance of context, namely, the co-occurrence of APOBEC activity with compromised repair capacity (e.g., low APE1) in a BRCA2-deficient background.

Recent work[125] studying patient prognosis in high-grade serous ovarian cancer (HGSOC) finds that APOBEC high (high APOBEC mutational signature) correlates with poor patient prognosis, consistent with our findings. We also examined *BRCA2*-mutant breast cancer cases using the METABRIC dataset. While the same trend was observed, i.e., low APE1 levels correlated with poorer survival, it did not reach statistical significance in the breast cancer cohort. This discrepancy may reflect treatment differences, as *BRCA2*-mutant breast cancers are typically estrogen receptor-positive (ER +) and are treated with hormonal therapies rather than platinum-based agents or PARP inhibitors, which are standard first-line therapies in BRCA2-deficient ovarian cancer. Because APE1 loss modulates sensitivity to replication stress-inducing agents like cisplatin rather than hormonal therapies, its impact on survival may be less pronounced in BRCA2 mutant breast cancer settings.

Finally, while our experiments employed exogenous replication stress to reveal the A3B-driven genomic instability loop, BRCA2-deficient tumors inherently experience chronic, endogenous replication stress due to impaired fork protection and unresolved replication intermediates. Thus, the A3B-driven mutagenic program is likely active even in the absence of therapy and is further amplified by treatment. This work also raises intriguing questions about whether BRCA2-heterozygous mammary epithelial cells in mutation carriers, which have been shown to be haploinsufficient for replication stress suppression[126,127], may engage this self-reinforcing A3B loop early during tumorigenesis, thereby contributing to tumor evolution.

In conclusion, our findings reveal a unique and clinically relevant vulnerability in BRCA2-deficient cells – a self-amplifying mutagenic feedback loop involving APOBEC3B, UNG2, APE1, and cytoplasmic ssDNA fragments. This loop links replication stress to cytidine deamination, stalled fork collapse, A3B upregulation, and UNG2 and APE1-driven genomic instability. These insights deepen our understanding of how BRCA2-deficient tumors evolve, and offer alternate therapeutic opportunities, targeting A3B, UNG2, APE1, and the NF-κB axis, to suppress mutagenesis, delay resistance, and improve outcomes in *BRCA2*-mutant cancers.

## Methods

### Cell lines and cell culture

U2OS BICR6, HeLa, PEO1, PEOC4, U2OS A3B KO, and U2OS APOBEC3A/3B DKO cell lines were grown in DMEM supplemented with 10% of FBS, 100 μ/ml Penicillin-Streptomycin, and 2 mM L-Glutamine. U2OS UNG2 KO and U2OS TRIPZ shBRCA2 cells were additionally cultured with 2.5 μg/mL puromycin. shRNA expression in TRIPZ shBRCA2 cells was induced with 10 μg/mL doxycycline for 48 h. HeLa cell lines with or without FLAG-tagged APE1 complementation were maintained in blasticidin (3 μg/mL), zeocin (10 μg/mL), and geneticin (400 μg/mL). For siAPE1 induction, Hela cells were treated with 1 μg/mL doxycycline

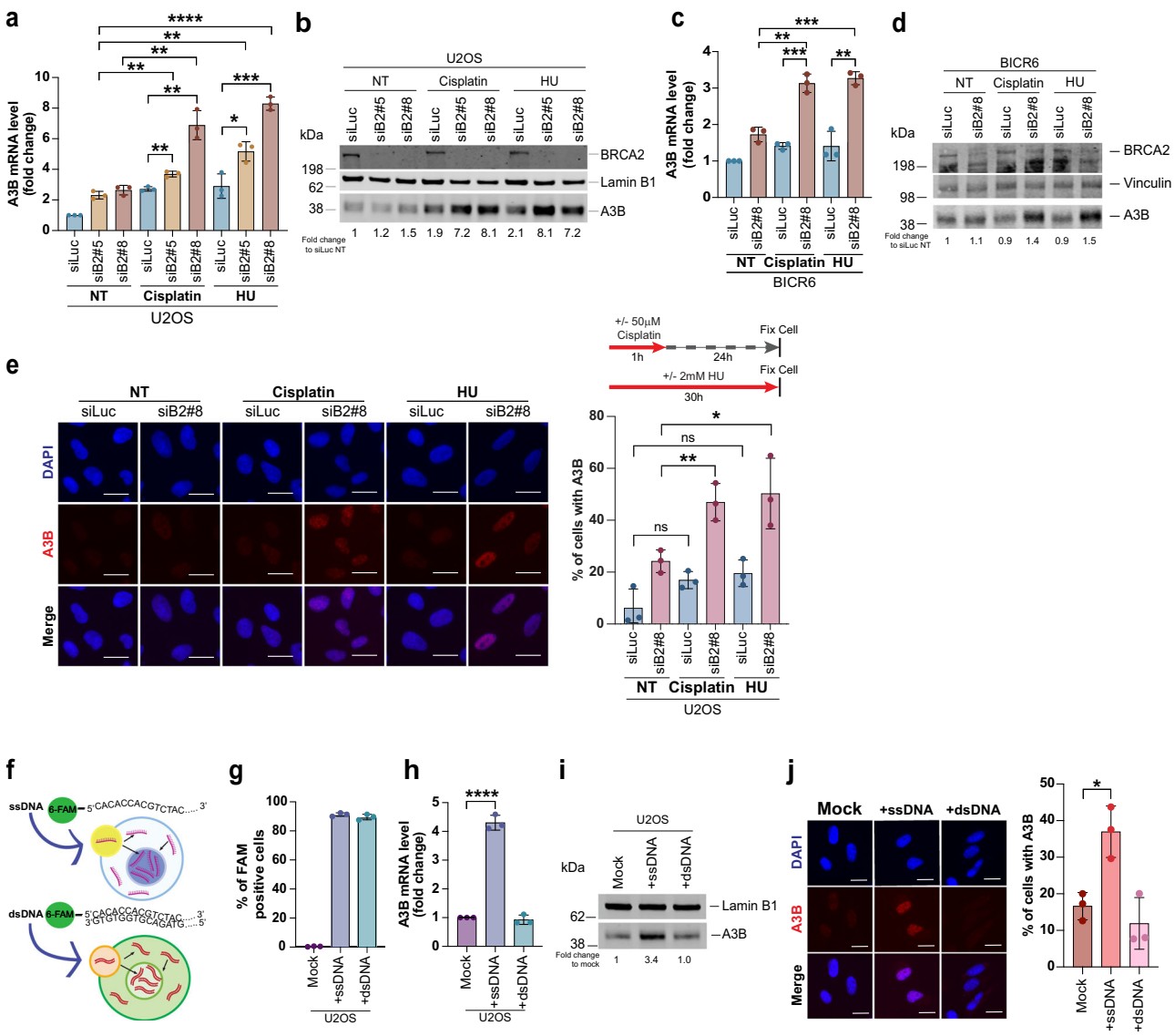

**Fig. 7 | A3B expression is induced by replication stress and ssDNA. a** qRT-PCR of A3B mRNA in U2OS cells transfected with indicated siRNAs and treated with cisplatin (50 µM, 1 h, 24 h recovery) or HU (2 mM, 30 h). Data is generated from $n = 3$ biological replicates; mean ± SD ($n = 3$). Statistical significance was determined by an un-paired two-tailed Student's $t$ test. ****$p \le 0.0001$; ***$p \le 0.001$; **$p \le 0.01$. **b** Western blot of BRCA2 and A3B in U2OS cells transfected with the indicated siRNAs and treated with cisplatin or HU as described in (**a**). LaminB1 is the loading control. One of three independent experiments is presented. A3B expression is normalized to loading control prior to fold change compared to siLuc untreated (NT). **c** qRT-PCR of A3B mRNA in BICR6 cells transfected with indicated siRNAs and treated with cisplatin or HU. Data presentation and statistical quantification is as described in (**a**). **d** Western blot of BRCA2 and A3B in BICR6 cells described in (**c**). Vinculin is the loading control. Fold change is calculated as described in (**b**). **e** Left: Representative images of A3B immunostaining in U2OS cells transfected with indicated siRNAs and treated with cisplatin or HU. Scale bar represents 20 µm.

(Right) Percentage of A3B-positive cells. Data are shown as mean ± SD of $n = 3$ independent experiments; > 200 cells/replicate. Statistical significance was determined by an un-paired two-tailed Student's $t$ test. **$p \le 0.01$; *$p \le 0.05$; ns, not significant. **f** Schematic of 6-FAM-tagged ssDNA or dsDNA (65nt). **g** FACS analysis of FAM-positive U2OS cells transfected with ssDNA or dsDNA (65nt) for 24 h. **h** qRT-PCR of A3B mRNA in U2OS cells transfected with 2.07 nM ssDNA or dsDNA for 24 h. Data are mean ± SD of $n = 3$ biological replicates. Statistical significance was determined as in (**e**). ****$p \le 0.0001$ (**i**) Western blot of A3B in nuclear extracts of U2OS cells treated as in (**h**). Lamin B1 is the loading control. Fold change is calculated as described in (**b**). **j** Left: Representative A3B immunostaining in U2OS cells transfected with mock, ssDNA, or dsDNA for 24 h. Scale bar represents 20 µm. Right: Percentage of A3B positive cells. Statistical significance was as described in (**e**). *$p \le 0.05$. Western blot images presented represent ≥ 3 western blots with similar results. Source data provided as a Source Data file.

for 10 days. HCT116 chr3+ (MMR competent) cells, with or without UGI, were cultured in McCoy's 5 A medium containing 10% FBS and antibiotics as described above. DLD-1 cells were cultured in RPMI with 10% of FBS, 2mM L-Glutamine, and antibiotics. CAPAN-1 and C2-12 cells were grown in IMDM with 20% FBS, 2mM L-Glutamine, and antibiotics. UWB1.289 cells were grown in a 1:1 mixture of RPMI and MEGM (Lonza CC-3150) supplemented with 3% FBS. U2OS (catalog #HTB-96) and UWB1.289 (catalog# CRL-2945) were purchased from American Type

Culture Collection (ATCC). U2OS WT (clone #1), U2OS A3B KO (clone #38), and U2OS APOBEC3A/3B (clone#29) KO cells were provided by Dr. Charles Swanton (Cancer Evolution and Genome Instability Laboratory, The Francis Crick Institute, London, UK) and generated as described in PMID: 33947663. PEO1, PEOC4[49], CAPAN1, and C2-12[111], cells were provided by Dr. Sharon Cantor (UMass Chan Medical School, Worcester, MA 01655 USA). U2OS TRIPZ shBRCA2 cells (Horizon Discovery Clone ID: V3THS-376145) were provided by Dr. Ryan

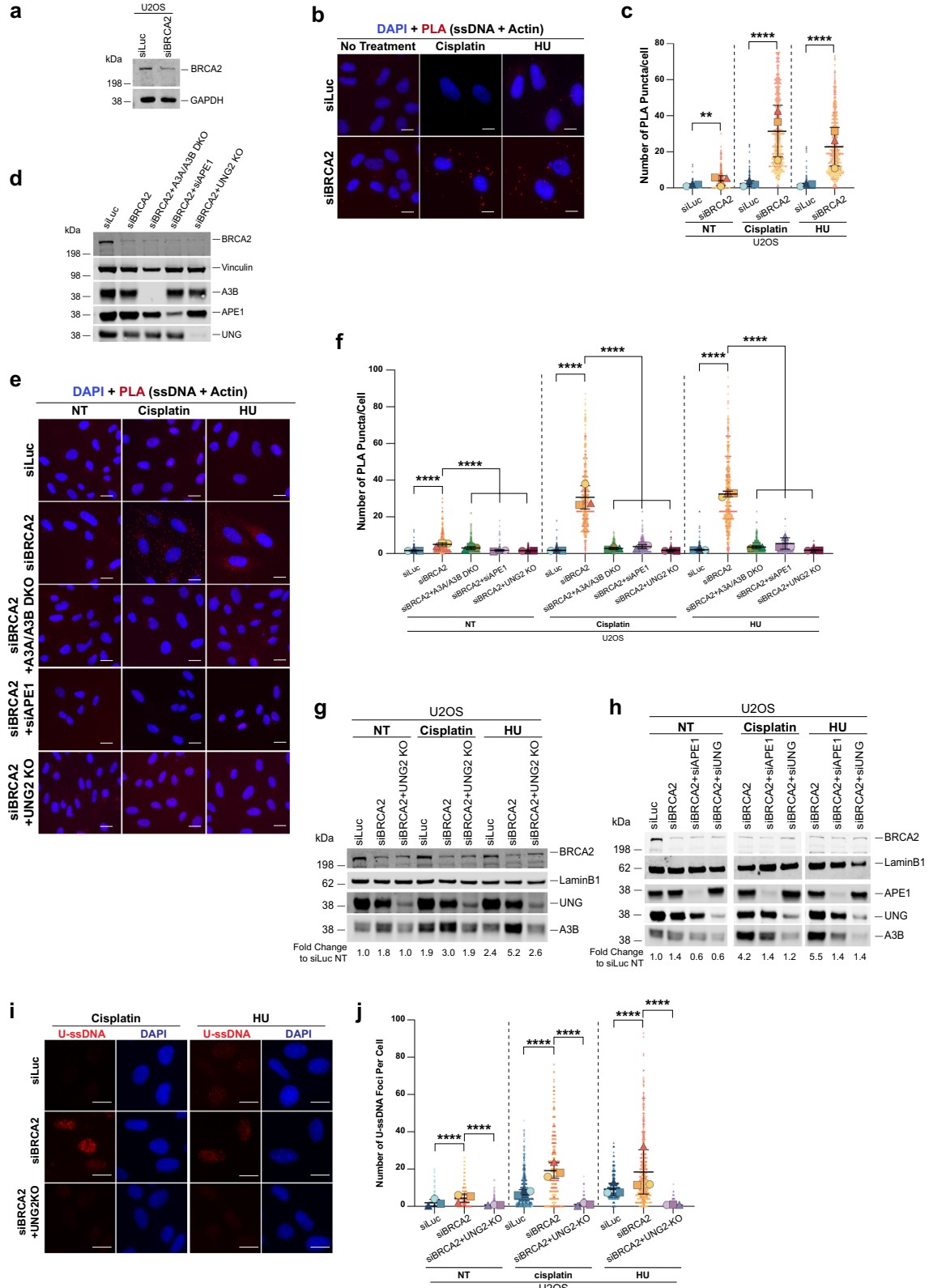

Jensen (Yale School of Medicine, New Haven, CT 06520 USA). HeLa cell lines containing doxycycline- inducible shAPE1 with or without flag-tagged APE1 complementation[128] was provided by Dr. Gianluca Tell (University of Udine, Udine UD, Italy). HCT116 chr3+ (MMR competent) with or without UGI cells[45] were provided by Dr. Beáta G Vértessy (BME Budapest University of Technology and Economics, Műegyetem Rkp.

3., Budapest, 1111, Hungary). DLD-1 (Catalog # CCL-221, ATCC) and DLD-1 BRCA2 knock out (Accession# CVCL_HD57) cells were provided by Dr. David Szuts (Institute of Enzymology, Research Center for Natural Sciences, 1117 Budapest, Hungary). BICR6 was provided by Dr. Abby Green (Washington University School of Medicine, St. Louis, MO, USA).

**Fig. 8 | BRCA2 deficiency induces accumulation of cytoplasmic ssDNA.**
**a** Western blot of BRCA2 in U2OS cells transfected with the indicated siRNAs. GAPDH is the loading control. **b** Representative images of proximity ligation assay (PLA) using anti-ssDNA and anti-actin antibodies in U2OS cells transfected with the indicated siRNAs and treated with cisplatin (50 μM, 1 h, 24 h) or HU (2 mM, 30 h). Scale bar represents 20 μm. **c** Quantification of PLA puncta/cell from (**b**). SuperPlots of three independent experiments are presented (n = 3). The number of PLA puncta/cell was analyzed through Image J; 100–200 cells/replicate. Each highlighted shape (circle, triangle, or square) represents the average of each replicate, with the black lines representing the mean ± SD. **d** Western blot of BRCA2, A3B, APE1, and UNG2 in U2OS cells used in (**e**, **f**). **e** Representative PLA images in U2OS cells transfected with the indicated siRNAs and then treated with cisplatin or HU. Scale bar represents 20 μm. **f** Quantification of PLA puncta/cell for (**e**). SuperPlots of

three independent experiments; 100–200 cells/replicate are plotted as described in (**c**). **g, h** Western blots of BRCA2 and A3B in nuclear extracts from wildtype or UNG2 KO U2OS cells transfected with the indicated siRNAs and treated with cisplatin or HU. LaminB1 is the loading control. Fold change compared to siLuc untreated (NT) is determined by first normalizing to the loading control.
**i** Representative U-ssDNA images in U2OS or UNG2 KO U2OS cells transfected with the indicated siRNAs and treated with cisplatin or HU. Scale bar represents 50 μM.
**j** Quantification of U-ssDNA foci. SuperPlots of three independent experiments are plotted as described in (**c**). Statistical significance for all charts presented here was determined by using a repeated measurement model followed by two-tailed multiple comparisons with a Bonferroni Post hoc test. ****$p \leq 0.0001$; **$p \leq 0.01$. Western blot images presented here are representative of three or more western blots with similar results. Source data are provided as a Source Data file.

## siRNA Transfection

Small interfering RNA (siRNA) transfections were carried out with Lipofectamine RNAiMAX reagent (Invitrogen, #13778150) based on the manufacturer's instructions. The sequences of the siRNAs used in this study are provided in the key resources table. Cells were transfected with 60 pmol siRNA, andthe medium was replaced the following day. Experiments and/or exposure to replication-stalling agents were carried out 48 h post siRNA transfection. Sequences used in this study are listed in Supplementary Table 1.

## Plasmid transfection

U2OS cells were transfected with plasmids using Lipofectamine 3000 (Invitrogen, #L3000015) according to the manufacturer's instructions. Plasmid transfection was carried out 24 h after siRNA transfection, and experiments or treatments with replication-stalling agents were carried out 48 h after plasmid transfection. Plasmids used in this study are listed in Supplementary Table 5.

## Generation of APE1 siRNA resistant plasmid

Human APE1/APEX1 cDNA with N-Myc tag (Sino Biological, #H616161-NM) was made siRNA resistant to siAPE1#2 using the QuikChange Multi Site-Directed Mutagenesis Kit (Agilent, #200514), following the manufacturer's instructions. Mutagenic primers (primer name: siAPE1#2_siRNA Resistance Mutagenesis, sequence listed in Supplementary Table 3) were used for site-directed mutagenesis, and the resulting PCR product was digested with *Dpn* I and transformed into XL10 Gold ultracompetent cells. Colonies were selected on ampicillin plates, and plasmids were confirmed using DNA Sanger sequencing.

## ssDNA and dsDNA Transfection

ssDNA oligonucleotides and their complementary strands were synthesized by Integrated DNA Technologies (IDT). To generate dsDNA, complementary oligonucleotides were annealed at 95 °C for 5 min and cooled to room temperature at −3 °C per 5 seconds. Cells were transfected with 2.07 nM ssDNA or dsDNA using Lipofectamine 3000 (Invitrogen #L3000001) according to the manufacturer's protocol. Cells were washed and collected 24 h post-transfection for protein isolation. Oligonucleotide sequences are listed in Supplementary Table 4.

## CRISPR Knock out cells

UNG2 CRISPR-Cas9 Double Nickase plasmid was purchased from Santa Cruz Biotechnology (SCBT, (sc-403189-NIC) and transfected into U2OS cells using Lipofectamine 3000 reagent (Invitrogen). After 3 days, GFP-positive cells were sorted into 96-well plates as single cells by FACS and then were selected with puromycin (2.5 μg/ml). Single-cell clones were validated by DNA Sanger sequencing. U2OS A3B KO or A3A/A3B DKO cells were created as described previously[33]. KO cell lines were further confirmed by DNA Sanger sequencing.

## DNA damaging agents and inhibitors

Cisplatin (50 μM; Selleckchem) was added 48 h post-siRNA transfection for 1 h, followed by 24 h recovery. Hydroxyurea (HU; 2 mM, Sigma-Aldrich) was added 48 h post siRNA transfection for 30 h without recovery. For micronuclei experiments, cells were allowed 48 h (cisplatin) and 24 h (HU) recovery before collection or fixation. For 53BP1 experiments, cells were treated with 5 mM HU for 4 h followed by 18 h of recovery. NF-κBi (10μM; Selleckchem BMS-345541) was added 48 h post siRNA transfection for 2 h and maintained during HU treatment and recovery. 5-fluoro-2′-deoxyuridine (5FdUR; 20 μM; Fisher Scientific) was added for 48 h. Methyl methanesulfonate (MMS, 10 mM, Sigma-Aldrich, 129925) was added to cells for 2 h.

## FLAG-ΔUDG constructs and purification

Catalytically inactive U-DNA sensor proteins (3xFLAG-ΔUNG) were purified as described previously[46]. Briefly, plasmids were transformed into *E. coli* Rossetta (DE3) (Millipore) and selected with chloramphenicol and ampicillin. A single colony was cultured overnight in LB medium at 37 °C, then expanded into 500 mL LB and grow until OD600 reached between 0.6–0.8. Protein expression was induced with 0.6 mM IPTG and continued overnight at 18 °C. Cells were harvested and lysed, and the supernatant collected after centrifugation. The supernatant was incubated with Ni-NTA resin (Invitrogen #R90115) at 4 °C for 1 h. Bound protein was purified using a series of low to high-salt buffers. The eluted protein was dialyzed overnight at 4 °C to remove imidazole, concentrated using a centrifugal filter (Millipore UFT801024), and quantified by spectrophotometer. Protein expression and purity were confirmed by western blotting.

## U-DNA detection assay

Uracil-in-DNA was detected using a 3 x FLAG-ΔUNG sensor as described in Pálinkás et al.[45] with minor modifications. Cells on coverslips were permeabilized with 0.2% TritonX-100 in CSK buffer (10 mM PIPES pH 6.8, 100 mM NaCl, 300 mM sucrose, 1 mM EGTA, 3 mM MgCl$_2$) for 2.5 min and then fixed in 4%PFA/2% sucrose solution for 10 min. For DNA denaturation, cells were treated with 2 M HCl at room temperature for 15 min followed by neutralization with 0.1 M Na$_2$B$_4$O$_7$ (pH 8.5) for 5 min. For uracil detection on ssDNA, DNA denaturation with HCL was omitted. Cells were blocked in Buffer I (50 mM Tris-HCL pH 7.4, 2.7 mM KCl, 137 mM NaCl, 0.05% Triton X-100, 5% non-fat dry milk) for 15 min, followed by incubation in Buffer I supplemented with 200 μg/ml salmon sperm DNA for 45 min at room temperature. Cells were then incubated with 4 μg/ml 3xFLAG-ΔUNG sensor in the same buffer for 1 h at room temperature. After washing twice with PBS, cells were incubated with anti-FLAG M2 antibody (1:10000, F1804, Sigma) followed by anti-mouse secondary antibody (1:400, 115-295-166, Jackson ImmunoResearch) for 1 h each at room temperature in blocking Buffer II (5% goat serum, 3% BSA, 0.05% TritonX-100 in PBS). Coverslips were mounted with DAPI.

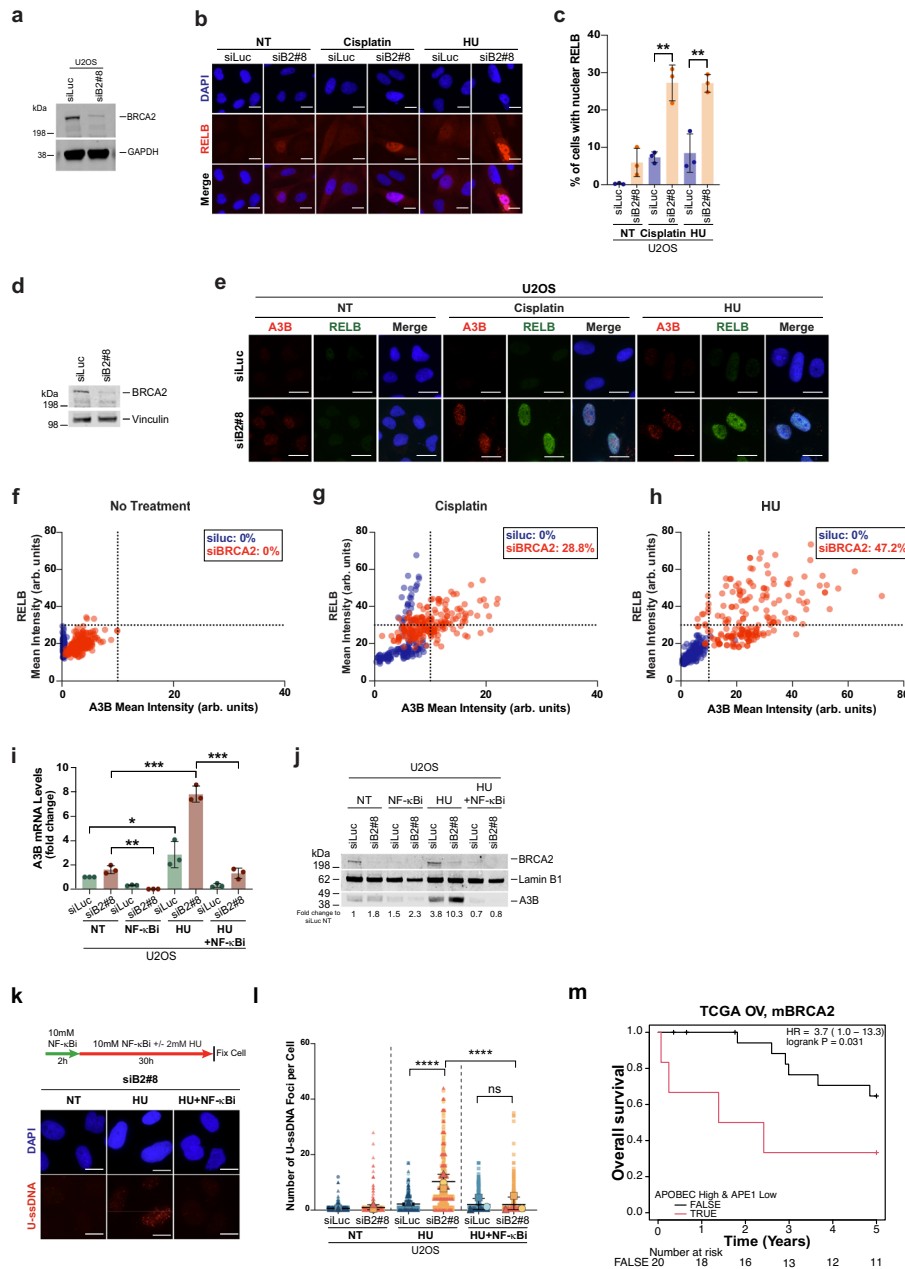

**Fig. 9 | NF-κB signaling drives increased APOBEC3B expression upon replication stress in BRCA2-deficient cells. a** Western blot of BRCA2 in cells transfected with the indicated siRNAs. GAPDH is the loading control. **b** Representative immunostaining and (**c**) quantification of nuclear RELB in U2OS cells transfected with the siRNAs and treated with cisplatin (50 μM, 1 h, 24 h recovery) or HU (2 mM, 30 h). Scale bar represents 20 μm. Data shown as mean ± SD of three independent experiments (*n* = 3). Statistical significance determined by un-paired two-tailed Student's *t* test. \*\**p* ≤ 0.01. **d** Western blot of BRCA2 in cells transfected with the indicated siRNAs. **e** Representative A3B and RELB co-immunostaining in U2OS cells transfected with indicated siRNAs and treated with cisplatin or HU. Scale bar = 20 μm. **f**–**h** Quantification of A3B and RELB mean intensity/cell in untreated (NT) (**f**), cisplatin-treated (**g**), or HU-treated (**h**) cells. Dashed lines indicate the baseline intensity (no-treatment condition). The percentage of cells with a mean intensity higher than baseline are indicated in the quadrant. **i** qRT-PCR of A3B mRNA in cells transfected with indicated siRNA and treated with or without NF-κBi (10 mM, 2 h pre-treatment, continued during HU exposure). Data shown as mean ± SD of *n* = 3

independent experiments. Statistical significance was determined as in (**c**). \*\*\**p* ≤ 0.001; \*\**p* ≤ 0.01; \**p* ≤ 0.05. **j** Western blot of BRCA2 and A3B in cells transfected with the indicated siRNAs. LaminB1 is the loading control. **k** Top: Schematic treatment with and without NF-κBi and HU. Bottom: Representative U-ssDNA images and (**l**) quantification of U-ssDNA foci in U2OS cells transfected with indicated siRNAs and pre-treated with NF-κBi (10 mM, 2 h) followed by HU (2 mM, 30 h). Scale bar represents 20 μm. >100 cells/replicate. SuperPlots of three independent experiments are presented. Each highlighted shape (circle, triangle, or square) represents the average/replicate, with black lines representing the mean ± SD. Statistical significance was determined by using a repeated measurement model followed by two-tailed multiple comparisons with a Bonferroni Post hoc test. \*\*\*\**p* ≤ 0.001; ns: not significant *p* > 0.05. **m** Kaplan-Meier curve of 5-year survival in the TCGA ovarian cohort with BRCA2-deficient tumors. Patients stratified by high APOBEC3A/B expression and low APE1 (below median). Associated BRCA2 patient mutations and source data are listed in the Source Data file. Western blot images are representative of ≥ 3 western blots with similar results.

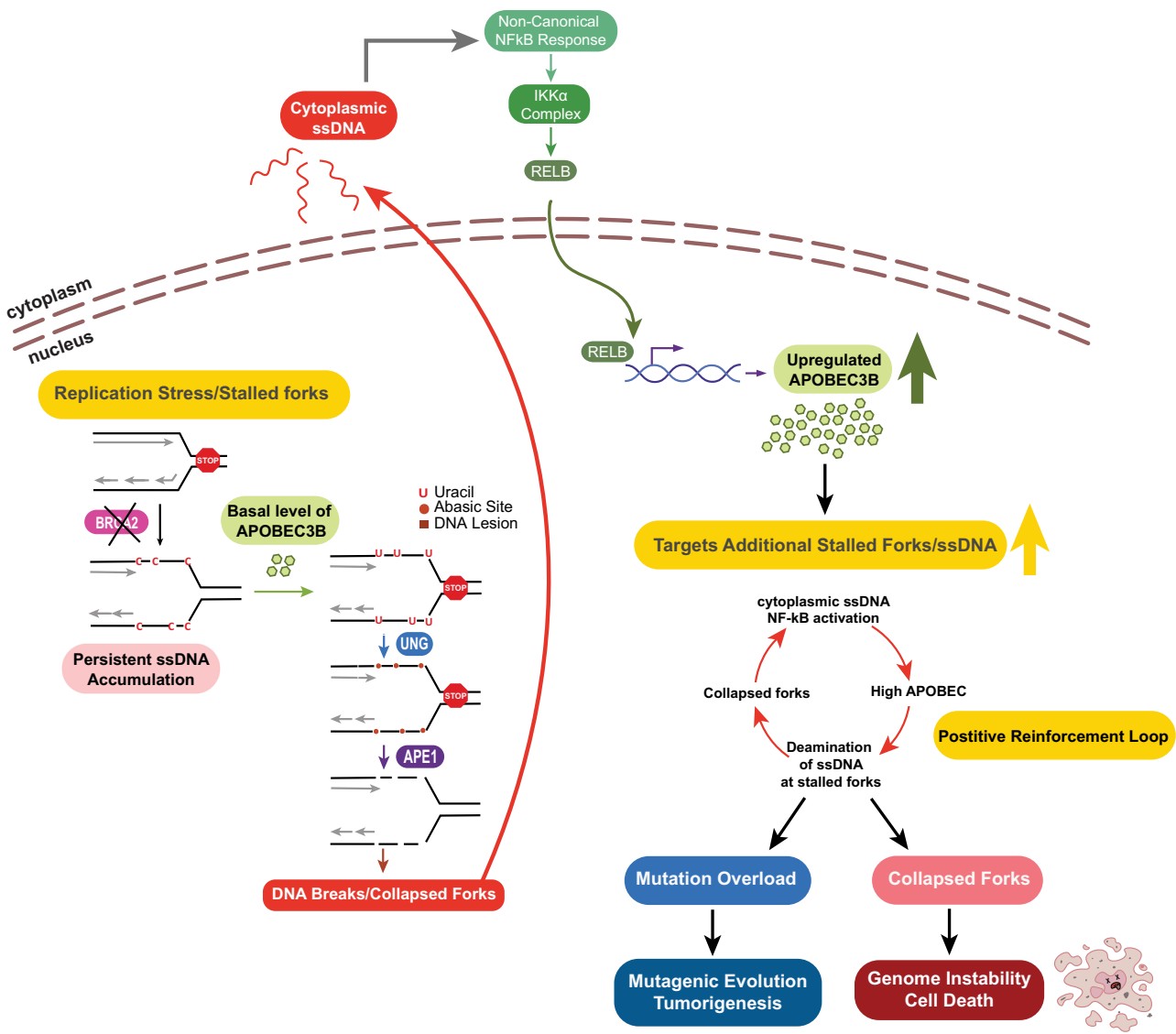

**Fig. 10 | Proposed model of APOBEC3B-driven mutagenesis and genomic instability in BRCA2-deficient cells.** Replication stress leads to stalled fork collapse in an A3B-, UNG2-, and APE1-dependent manner, resulting in the generation of ssDNA fragments in the cytoplasm which go on to activate NF-κB. This, in turn, drives APOBEC3B upregulation and establishes a positive self-reinforcing loop that promotes mutagenic evolution and genomic instability in BRCA2-deficient cells.

For S1 nuclease treatment, CSK-permeabilized cells were incubated with 20 μ/ml S1 nuclease (Invitrogen, 18001016) in S1 buffer (30 mM sodium acetate, pH 4.6, 1 mM zinc acetate, 5% (v/v) glycerol, 50 mM NaCl) for 15 min at 37 °C before fixation and immunostaining without HCL denaturation. For RNAse A treatment, cells were fixed (4% PFA/2% sucrose solution) after permeabilization and treated with 100 μg/ml of RNAse A (Invitrogen, #12-091-021) for 4 h at 37 °C, followed by blocking in 5% BSA and immunostaining without HCl denaturation.

For U-DNA detection with PCNA, CSK-permeabilized cells were incubated with 100% ice-cold methanol (15 min, 4 °C). Cells were blocked in Buffer I and incubated with 3xFLAG-ΔUNG probe for 1 h at room temperature. After washing twice with PBS, cells were incubated with anti-FLAG M2 (1:10000, F1804, Sigma) and anti-PCNA (1:400, Abcam #ab2426) antibodies, followed by incubation with secondary antibody for 1 h each at room temperature in blocking Buffer II. Coverslips were mounted with DAPI.

## Cell viability assay
2000 cells/well were seeded in triplicate in 96-well plates. Cells were treated with cisplatin or HU for 4 days, then allowed to recover in drug-free medium for two additional days. Cell viability was assessed using CellTiter-Glo reagents (Promega; G7572). Luminescence signal was measured using a BMG Labtech luminometer.

## Immunoblotting
Whole-cell extracts were prepared by lysis in NETN 450 buffer (450 mM NaCl, 20 mM Tris-HCl, pH 8, 0.5% NP-40, 1 mM EDTA) for 30 min at 4 °C. For nuclear extracts, cells were incubated in Protein Extraction Buffer (PEB; 0.5% Triton X-100, 20 mM HEPES pH 7, 100 mM NaCl, 3 mM MgCl$_2$, 300 mM sucrose) on ice for 20 min, followed by centrifugation at 5000 rpm for 10 min to remove the cytoplasmic fraction. Cell pellets were lysed in NETN 400 buffer (400 mM NaCl, 20 mM Tris-HCl, pH 8, 0.5% NP-40, 1 mM EDTA) for 1 h at 4 °C to obtain nuclear extracts. All buffers were supplemented with protease inhibitor and phosphatase inhibitor. Proteins were denatured at 80 °C for 10 min, resolved by 4–12% Bis-Tris SDS-PAGE, and transferred to 0.45 μm nitrocellulose membrane. Membranes were blocked with the Intercept (PBS) Blocking Buffer (Licor, 927-70001). The primary and secondary antibodies used in this study are provided in the key resources table. Membrane was imaged using the Licor Odyssey Instrument.

## Antibodies

The antibodies used in this study are listed in Supplementary Table 2.

## RNA isolation and quantitative PCR (qRT-PCR)

Total RNA was was extracted using QIAzol reagent (Qiagen) according to the manufacturer's instructions. RNA concentration and purity was checked by spectrophotometry (A260/280 ratio ≥1.8). cDNA was generated from 1 μg of RNA using iScript reverse transcription super-mix (Biorad). Quantitative PCR was performed using SYBR Green Fast qPCR mix (Abclonal) on a Quant Studio 3 qPCR machine (Applied Biosystems) using 100 ng of cDNA per reaction. Primers used in this study were synthesized at Integrative DNA Technologies (IDT) or Eton Bioscience. Primer sequences used in this study are provided in Supplementary Table 3. Relative expression was normalized to human Actin. Statistical analysis was performed using Prism (Version 10.2.3, GraphPad Prism Software). Statistical significance for each experiment was determined by a two-tailed Student's $t$ test and represented as a mean ± SD as indicated in the figure legends. The data presented are representative of 3 independent data sets with technical triplicates. Primer sequences for qRT-PCR are listed in Supplementary Table 3.

## DNA Fiber assay

Stalled fork degradation was studied using the DNA Fiber assay as described previously[129]. Briefly, cells were sequentially labeled with 25 μM IdU for 30 min and 250 μM CldU for 30 min, followed by PBS washes and treatment with 5 mM HU for 4 h. Unlabeled cells were mixed at a 1:2 ratio with labeled cells prior to lysis and spreading on to the tilted slides. DNA fibers were fixed in acetic acid:methanol (1:3) for 1 h at room temperature, denatured in 2 M HCl for 1 h at room temperature, and stained with rat anti-CldU (Abcam; ab6326) and mouse anti-IdU (BD Biosciences;555627) followed by staining with secondary antibodies Alexa Fluor 555 goat anti-rat (Thermo Fisher Scientific; A-21434) and Alexa Fluor 488 goat anti-mouse (Thermo Fisher Scientific; A-10684) for 1 h at room temperature for primary and secondary, respectively.

## Neutral and modified alkaline comet assay

Comet Assay was performed under neutral conditions using the CometAssay reagent kit (4250-050-KL, Trevigen). Briefly, $1 \times 10^5$ cells were embedded in LM Agarose (1:10) at 37 °C and immediately spread onto CometSlides (50uL/well). Slides were placed in 4 °C for 30 min to solidify the agarose, lysed overnight at 4 °C, washed in neutral electrophoresis buffer (100 mM Tris base, 300 mM sodium acetate, pH 9) for 1 h, and electrophoresed for 25 min at 21 volts. Slides were immersed in dH2O for 10 min before immersion in 70% ethanol for five minutes.

Modified UDG comet assays were performed under alkaline conditions. Briefly, after cell lysis, slides were washed in incubation buffer (40 mM HEPES, pH 8, 0.5 mM Na2EDTA, 0.2 mg/mL BSA, 0.1 M KCL) three times for 5 min at 4 °C[130]. Slides were treated with UDG (10 μ/mL) or buffer alone for 1 h at 37 °C in a humidified chamber. Enzymatic activity was stopped on ice, and slides were incubated in alkaline solution (pH > 13, 200 mM NaOH, 1 mM EDTA) for 20 min at room temperature before electrophoresis (30 min, 21 V). All steps were performed in the dark.

Slides from neutral and alkaline comet assay were dried, stained with SYBR Gold (1:10000, S11494, Invitrogen), and imaged on a Zeiss Axio Imager.M2 equipped with an Axiocam 506 color camera. Tail moment and Olive moment were quantified using ImageJ/OpenComet.

## Abasic Site (AP) labeling, and quantification

Cells ($1 \times 10^6$) were plated in a 60 mm dish 24 h after siRNA transfection and treated with hydroxyurea or cisplatin for the indicated times. Harvested cells were washed with PBS and incubated with 2.5 mM aldehyde reactive probe (ARP; Cayman 10009350) for 1 h at 37 °C with gently shaking. Genomic DNA was extracted using the QIAmp DNA Mini Kit (Qiagen 51306) and quantified by Nanodrop 2000 (ND-2000, Thermo Scientific). For dot blot analysis, 1μg of ARP-labeled DNA was applied to a positively charged nylon membrane (Zeta-Probe 1620157, BioRad) presoaked in 6x SSC (20x SSC: 3.0 M NaCl and 0.3 sodium citrate at pH 7.0) in a 96-well dot blot apparatus (1706545, BioRad). Membranes were washed with TE (10 mM Tris-HCL, 1 mM EDTA), rinsed with 2x SSC, dried, baked at 80 °C for 2 h, and blocked in Li-Cor blocking buffer (927-70001) for 1 h. Blots were probed with IRDye 800CW Streptavidin (926-32230, Li-Cor), imaged on Li-Cor Odyssey CLx, and quantified using Image Studio Lite (Li-Cor). AP-DNA signal was normalized to the methylene blue-stained DNA signal (MB119, Molecular Research Center). Statistical significance was assessed by a two tailed Student's $t$ test, with $p < 0.05$ considered significant. Data represent at least 3 independent data sets.

## Immunofluorescence staining

Cells grown on coverslips were treated with hydroxyurea or cisplatin for the indicated times. Coverslips were fixed with 4% PFA/2% sucrose (10 min), and permeabilized with 0.5% Triton X-100 in PBS (4 min, RT). Cells were washed with 1% BSA and incubated with primary antibodies at 37 °C for 30 min, followed by secondary antibodies. Coverslips were mounted with DAPI-containing mounting medium, and images were acquired with an Axio Imager.M2 microscope (Carl Zeiss) equipped with an Axiocam 506 color camera and Zen software.

For staining after pre-extraction, cells were washed twice with PBS and permeabilized with CSK buffer (100 mM NaCl, 300 mM sucrose, 10 mM PIPES, pH 7.0, 3 mM MgCl$_2$), followed by incubation with 0.2% Triton X-100 for 2.5 min at room temperature (RT). Cells were further washed with CSK buffer, then fixed with 4% PFA/2% sucrose in PBS (15 min). Cells were washed with PBST (0.1% Tween-20 in PBS), blocked with 5% BSA in PBST (15 min, RT), and incubated with primary antibodies (37 °C, 30 min) followed by secondary antibodies.

For ssDNA detection, cells were treated as above with PBS and CSK buffer before permeabilization with 0.2% Triton X-100 (2.5 min, RT). Cells were fixed with 4% PFA/2% sucrose in PBS (15 min) and further incubated with 100% ice-cold methanol (15 min, 4 °C). Following PBS washes, cells were blocked with 5% BSA (15 min, RT) and incubated with ssDNA antibody (Millipore Catalog #MAB3299) and β-Actin antibody (Cell Signaling # 4970S) at 37 °C for 30 min, followed by incubation with secondary antibody.

For A3B detection, cells were treated and fixed as above with PBS, CSK, and PFA. Cells were denatured with 2 M HCl at room temperature for 10 min followed by neutralization with 0.1 M Na$_2$B$_4$O$_7$ (pH 8.5) for 5 min. Following PBS washes, cells were blocked with 5% BSA (15 min, RT) and incubated with anti-A3B (1:500, Cell Signaling #41494) at 37 °C for 30 min followed by incubation with secondary antibody.

## Proximity ligation assay (PLA)

PLA was performed using the Duolink® In Situ Detection Reagents following the manufacturer's instructions. Cells grown on coverslips were treated with hydroxyurea or cisplatin, pre-extracted with CSK buffer and 0.2% Triton X-100, and fixed with 4% PFA/2% sucrose (15 min, RT), followed by incubation with 100% ice-cold methanol (15 min, 4 °C). Cells were blocked in Blocking Solution (Millipore DUO92001 or DUO92005) for 1 h at 37 °C in a humidity chamber. After blocking, cells were incubated with primary antibodies (anti-ssDNA: Millipore #MAB3299, anti- β-Actin: Cell Signaling #4970S) diluted in the Antibody Diluent and incubated for 30 min at 37 °C in a humidity chamber. Cells were washed twice for 5 min in Wash Buffer A at room temperature. PLUS and MINUS PLA probes, ligase, and polymerase were applied sequentially according to the kit protocol. Coverslips were mounted with DAPI-containing mounting media.

## Micronuclei analysis

Cells seeded on coverslips were treated with hydroxyurea or cisplatin for the indicated times and allowed to recover. Cells were fixed in 4% PFA/2% sucrose, permeabilized with 0.05% Triton X-100, and mounted with DAPI-containing mounting media. ≥ 400 cells per condition were counted manually.

## Colony formation assay

20,000 cells were seeded onto a 6-well plate prior to treatment with cisplatin or HU for 4 days and allowed to recover for 6–8 days. Cells were fixed and stained with 6.12 mM Crystal Violet (0.125grams/50 mL) containing 20% methanol. Plates were washed six times with water and dried. Plates were imaged using Licor Odyssey and quantified with ImageStudioLite2 as arbitrary units. An empty 6-well with no cells was used to determine 0% survival, and no treatment wells, respectively, were used to determine 100% survival using a built-in-analysis- 'Normalize' function in Prism (V. 10.2.3, GraphPad Prism Software). Percent colony survival was calculated using Prism (Version 10.2.3, GraphPad Prism Software).

## Image acquisition and analysis

Images were acquired on an Axio Imager.M2 (Carl Zeiss) equipped with an Axiocam 506 color camera and controlled by Zen software. X-Cite XT120L-Boost (Excelitas) was used as the light source for fluorescence. Micronuclei (≥ 200 cells/replicate) were scored by manual counting. Mean intensity within a defined nucleus was quantified using the "Measure" or "Define Particles" function in ImageJ. Uracil foci and PLA puncta were quantified in Image J using the "Find Maxima" function (prominence set to 30.0). Co-localization between U-ssDNA and PCNA was determined with the Colocalization.class plugin, followed by the "Find Maxima" function (prominence set to 25.00). For comet assays > 100 cells/replicate were measured for tail and olive moments using Image J/Open Comet. For DNA fiber analysis, 100–300 fibers were measured per condition using Image J.

## Quantification and statistical analysis

All quantitative data represent at least three independent experiments or biological replicates unless otherwise indicated. Statistical analysis was performed using Prism (Version 10.2.3, GraphPad Prism Software). Statistical significance for each experiment was determined by two-tailed Student's $t$ test and represented as a mean ± SD (standard deviation) or by repeated measurement ANOVA followed by multiple comparisons with Bonferroni Post hoc test as indicated in the figure legends. In all cases, $*p \leq 0.05$; $**p \leq 0.01$; $***p \leq 0.001$; $****p \leq 0.0001$; ns: not significant $p > 0.05$. For SuperPlots[131], a repeated measurement model was used to compare across biological replicates and to assess differences among different conditions/groups. When significant effects were detected, post hoc multiple comparisons are conducted to determine the pairwise significance using the Bonferroni correction methods. SPSS version 29.0 was used for the statistical analysis.

## TCGA OvCa data analysis

RNAseq data from ovarian cancer tumors were obtained from The Cancer Genome Atlas (TCGA), processed by the University of California, Santa Cruz (UCSC) Toil pipeline[132], and summarized to transcripts per million (TPM) on the gene level. BRCA2 mutation status was obtained from the cBioPortal for Cancer Genomics[133,134], using the "GERMLINE", "DRIVER", and "HOMDEL" annotations. BRCA2 deficiency was defined as any of above events affecting BRCA2. To define tumors with high and low values of APE1, tumors were split by the median expression value in the BRCA2-deficient subset. To estimate APOBEC levels, the average of the log2 normalized TPM value of the APOBEC3A and APOBEC3B genes were used. Tumors were split into high and low by the median value. A list of mutations associated with the patient data used in this study are provided in the source data.

## Reporting summary

Further information on research design is available in the Nature Portfolio Reporting Summary linked to this article.

## Data availability

All data, materials, cell lines, and other reagents are available from the corresponding author upon request. All data supporting the findings of this study are available within the paper and its Supplementary Information. Source data are available in Figshare with the identifier https://doi.org/10.6084/m9.figshare.29604653. Source data are provided in this paper.

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

## Acknowledgements

This work was supported in part by NIH grants R01CA273696 and R15CA235436 (to S.P.), NIH Tufts IRACDA Partner Faculty Research Award Program (to S.P.), the Department of Defense Breast Cancer Program grant BC160079 (to S.P.), Basser Research Center for BRCA (to S.P.), Goldberg Grant (to S.P.), UMB College of Science and Mathematics Dean's Doctoral Research Fellowship (to H.D and K.S.), Alton J. Brann Endowment Undergraduate Research funds at UMB (to G.Q.M., B.N., S.S., and P.R.), National Research, Development and Innovation (NRDI) Fund of Hungary (B.G.V.: K135231, K-146890, A.B.: FK137867), TKP2021-EGA-02 grant by the Ministry for Innovation and Technology of Hungary from the NRDI Fund (to B.G.V.), János Bolyai Research Scholarship of the Hungarian Academy of Science (BO/726/22/8) (to A.B.), Francis Crick Institute core funding from Cancer Research UK (CC2041), the UK Medical Research Council (CC2041), and the Wellcome Trust (CC2041) Cancer Research UK (TRACERx (C11496/A17786), PEACE (C416/A21999) and CRUK Cancer Immunotherapy Catalyst Network); Cancer Research UK Lung Cancer Center of Excellence (C11496/A30025); the Rosetrees Trust, Butterfield and Stoneygate Trusts; NovoNordisk Foundation (ID16584); Royal Society Professorship Enhancement Award (RP/EA/180007 & RF\ERE\231118); National Institute for Health Research (NIHR) University College London Hospitals Biomedical Research Center; the Cancer Research UK-University College London Center; Experimental Cancer Medicine Center; the Breast Cancer Research Foundation (US) (BCRF-23-157); Cancer Research UK Early Detection an Diagnosis Primer Award (Grant EDDPMA-Nov21/100034); and The Mark Foundation for Cancer Research Aspire Award (Grant 21-029-ASP) and ASPIRE Phase II award (Grant 23-034-ASP), ERC Advanced Grant (PROTEUS) from the European Research Council under the European Union's Horizon 2020 research and innovation program (grant agreement no. 835297) (to C.S.). For the purpose of Open Access, C.S. has applied a CC BY public copyright license to any Author Accepted Manuscript version arising from this submission. We thank Ryan Jensen (Yale School of Medicine, New Haven, CT) for the TRIPZ shBRCA2 U2OS cells, Gianluca Tell (University of Udine, Udine UD, Italy) for the HeLa doxycycline induced shAPE1 cells with APE1 complementation, Dávid Szüts (Hungarian Academy of Sciences, Budapest, Hungary) for DLD-1 and DLD-1 BRCA2[-/-] cells, Sharon Cantor (UMass Chan Medical School) for PEO1/PEOC4 and CAPAN1/C2-12 cells, Bodil Kavli (Norwegian University of Science and Technology, Norway) for the p_UNG2_EYFP plasmid and Angela Gronenborn (University of Pittsburg, Pittsburg, PA) and Judith Levin (NIH, Bethesda, MD) for the A3B overexpression plasmid. We thank Ahkil Adusumilli for his contributions to the analysis of the U-ssDNA foci, and Zihan Li from the Department of Statistics at UMB for his assistance with the statistical analysis.

## Author contributions

K.S., H.D., and S.P. conceived the study and designed the experiments. K.S. and H.D. carried out experiments and analyzed the data along with S.P. Additional experiments were carried out by S.K.G. J.Y., M.K.G., G.Q.M., M.H.Z., S.S., P.R., N.P., B.M., and S.K.M. B.G.V., A.B., and H.P. provided the uracil-DNA probe (3XFLAG-ΔUNG) constructs and guided the U-ssDNA studies. C.S. and S.V. provided the A3B and A3A/A3B DKO U2OS cells. A.G. shared the A3B qPCR primers and A3A and A3B (wildtype and mutant) overexpression plasmids. Patient data and cBioPortal-based statistical analysis was carried out by N.J.B. Statistical analysis was carried out by J.C. During the course of this study, S.K.G., B.G.V., A.B., H.P., C.S., S.V., and A.G. participated in helpful discussions and helped with editing of the manuscript. The manuscript was written, edited, and prepared by K.S., H.D., and S.P.

## Competing interests

The authors declare no competing interests.

## Additional information

[1]Center for Personalized Cancer Therapy, University of Massachusetts Boston, Boston, MA, USA. [2]Department of Biology, University of Massachusetts Boston, Boston, MA, USA. [3]The Center of Statistical Computing, University of Massachusetts Boston, Boston, MA, USA. [4]Department of Applied Biotechnology and Food Sciences, Faculty of Chemical Technology and Biotechnology, BME Budapest University of Technology and Economics, Műegyetem Rkp. 3, Budapest, Hungary. [5]Institute of Molecular Life Sciences, Research Centre for Natural Sciences, Hungarian Research Network, Budapest, Hungary. [6]Cancer Research UK Lung Cancer Centre of Excellence, University College London Cancer Institute, London, UK. [7]Department of Pediatrics, Washington University School of Medicine, St. Louis, MO, USA. [8]Department of Molecular Medicine, Aarhus University Hospital, Aarhus, Denmark. [9]Department of Clinical Medicine, Aarhus University, Aarhus, Denmark. [10]Cancer Evolution and Genome Instability Laboratory, The Francis Crick Institute, London, UK. [11]Department of Oncology, University College London Hospitals, London, UK. [12]These authors contributed equally: Kathy Situ, Haohui Duan. ✉e-mail: Shailja.pathania@umb.edu

