## [Transparent Peer Review file · Nature Communications]

BRCA2 Deficiency and Replication Stress Drive APOBEC3-Mediated Genomic Instability

Corresponding Author: Dr Shailja Pathania

Version 0:

Reviewer comments:

Reviewer #1

(Remarks to the Author)

This manuscript uncovers a role for APOBEC activity as a key mediator of chemotherapy response in BRCA2-mutant cells. This work can have an important contribution in our understanding of mechanisms underlying genomic instability and drug response in BRCA2-deficient cancers. Many findings reported in this manuscript are interesting but at the same time unanticipated. The reviewer therefore strongly emphasizes on addressing the conceptual gap and inclusion of key control experiments (discussed below) to rigorously validate the observations and hence potentially avoid any misleading conclusions.

1. A key conceptual gap in this paper is the underlying source of single-strand DNA that distinguishes the phenotype observing between BRCA1- and BRCA2- mutant cells. Functions of BRCA1 and 2 are overall overlapping in the context of reversed fork or gap protection. In contrast, in the context of DSB repair, BRCA1 promotes resection and both BRCA1/BRCA2 contribute to Rad51 loading. Given that authors are observing increased accumulation of DSBs (Fig.5a-c), one likely explanation can be that resected DNA at collapsed forks can account for the phenotype in BRCA2-deficient cells. In contrast, lack of resection in BRCA1-deficient cells suppresses the phenotype. One approach could be testing a BRCA1 mutant (studied extensively by Neil Johnson's lab) proficient in resection but deficient in Rad51 loading.

2. Critical need for inclusion of important controls:

(i) It is imperative that authors demonstrate that Uss-DNA is enriched in S-phase. Differences being observed can be a simple manifestation of alterations in cell cycle upon Cisplatin or HU treatment.

(ii) AP DNA quantification: The signal needs to be normalized with the input DNA.

(iii) Establish the specificity of APEX1 siRNAs by either complementing back cDNA or citing its validity from previous studies.

(iv) S1 nuclease experiment: S1 nuclease digests both single-strand DNA and RNA. Therefore, utilizing this nuclease to conclude that Uracil is enriched on ss-DNA is misleading. Perhaps complementing the experiment with RNase A digestion which specifically degrades RNA at C or U residue will be more meaningful.

(v) Fork degradation experiment: The scheme used for this experiment is not logical. Authors use IdU followed by HU and then CldU. They measure the IdU track as a marker of fork degradation. This scheme assumes that all IdU tracks started at the same time and neglects the fact that origins don't fire synchronously. Hence the difference being observed can be a mere manifestation of IdU track starting at different time points. A more established way to address fork degradation is IdU, followed by CldU and then HU. Then measure the length of CldU tracks attached to IdU. This way one can ensure that all CldU tracks start at the same time. It is imperative that a head-to-head comparison is performed with BRCA1-mutant cells. Based on established literature one would expect similar results in BRCA1-settings. This would further help parse out if the source of ss-DNA is simply forking collapse or resected fork.

(vi) Use of 53BP1 (and that too without any cell cycle marker) as a read out of fork collapse is truly misleading. As authors would agree that 53BP1 marks various lesions and hence using it as a marker of replication fork collapse is incorrect. This reviewer suggests substituting 53BP1 experiment with fork degradation experiments as discussed above.

(vii) Fig.5m and Supp 5l. Were all BRCA1 and 2 patient-mutation analyzed established to be pathogenic? If not, one limitation of this analysis can be that many mutations were benign and hence no correlation was observed in the context of

BRCA1. It's important that the authors note this limitation in the manuscript.

(viii) Fig.6k. The finding that A3A/A3B dKO can rescue Cisplatin sensitivity of BRCA2-mutant cells is very striking and unexpected. Typically, A3A and A3B are negligibly expressed in most cell lines and hence assuming that these cytidine deaminases actually contribute to the majority of cisplatin sensitivity is surprising. U2OS cells are known to express A3A and A3B. It is imperative that authors validate these findings in relevant BRCA2-mutant lines as Pe01 and CAPAN-1 to avoid any misleading conclusions.

(ix) Figure 8d: Why is there an enrichment of U-DNA foci in siLuc, HU+ NFKBi samples. Also, to make the conclusion that NFKBi reduces U-DNA foci, the correct statistical comparison should be between HU treated vs HU+ NFKBi treated siBrca2 samples.

Minor points

1. Please include the schematic of treatment and the cell line names in the figures for the ease of readability.

2. Missing Citations:

Studies showing direct evidence that APE1 can function at the replication fork and introduce breaks: PMID: 39605722 and 39068174

Reviewer #2

(Remarks to the Author)

BRCA2 is important for the protection of stalled/regressed forks upon replication stress induction and in the absence of BRCA2, forks undergo premature degradation and collapse. In this manuscript, Duan et al demonstrate that the genomic instability observed in BRCA2-deficient cells upon replication stress induction is majorly driven via cytosine deamination by APOBEC3B (A3B). The authors have used creative assays to show that cisplatin or HU treatment in the absence of BRCA2 causes increased uracil accumulation in ssDNA via APOBEC3B-mediated cytosine deamination. Excision of uracils by UNG2 generates AP sites which are then cleaved by APE1 to induce breaks which are normally repaired via homologous recombination pathway. In the absence of BRCA2, HR is compromised causing unrepaired DSBs, genomic instability and cell death. The authors further propose that replication stress induction in BRCA2-deficient cells leads to increased A3B expression via activation of the NFKB pathway. This creates a feed-forward loop leading to more deamination and more breaks. Overall, it's a sound manuscript that brings forward interesting ideas in the APOBEC field. Most conclusions presented are clear and have strong evidence in their support.

Following few comments could make the manuscript stronger and more robust:

Major points:

1. Given that UNG is the primary glycosylase to excise uracils from DNA, it is surprising to see strong uracil signals from the U-ssDNA assay even in the presence of UNG in Figures 1 and 2. The authors also show that cells have increased AP sites in the presence of UNG which reduce after UNG KO in Fig. 3. The authors should ideally show appropriate controls for this assay by adding UNG KO panel. UNG KO experiments have been performed for other assays in subsequent figures but not for the critical U-ssDNA assay.

2. As the authors point out, treatments with cisplatin and HU would cause fork reversal. During fork reversal, nascent DNA strands anneal together limiting the amount of exposed ssDNA. As a result, majority of the cytosines would be end up in dsDNA which cannot be deaminated by A3B. Even if the deamination of cytosines to uracils happened when they were in ssDNA and then forks were reversed, upon reversal, the uracils would be in dsDNA and should not be detectable by U-ssDNA assay.

3. The ssDNA in BRCA2-deficient cells after cisplatin or HU would be quickly protected by RPA. How does RPA bound ssDNA get deaminated by APOBEC3B?

4. Fork degradation in BRCA2-deficient cells is via MRE11/EXO1-mediated exonuclease activity. In Supplementary figure 4, why does loss of UNG or APE1 i.e the increased presence of uracils or AP sites affect exonuclease activity of MRE11/EXO1? Similarly, In Supplementary Figure 6i, the fork degradation phenotype is completely reversed by A3B loss. These results from the fork degradation assays are a bit puzzling.

As a subpoint to that, the labeling scheme needs to be modified. Based on the information in the legend, there should be a horizontal black line showing HU treatment of 3hrs in between IdU and CldU. Moreover, if only IdU tracts are being measured, what is the reason to add CldU? Are the authors measuring fork degradation, stalled forks, fork restart?

5. Increased replication stress in BRCA2-deficient cells after cisplatin and HU treatment caused increased A3B expression. The authors make a claim that it is due to increased export of DNA in the cytoplasm leading to the activation of NFKB pathway. However, no direct experiment provides evidence for this claim. Transfected ssDNA leads to A3B activation (Fig. 7h-m) and RELB nuclear translocation (Fig. S7C) but that is completely different from nuclear DNA being released into the cytoplasm after replication stress causing NFKB activation and increased A3B expression. To solidify these conclusions, the authors need to perform several key experiments.

- The authors should show IF images showing A3B and RELB activation in the same nuclei after cisplatin or HU treatment in BRCA2-deficient cells.

- PLA assay with EdU (nascent nuclear DNA) and actin (cytosolic marker) to show increased cytosolic DNA after cisplatin or HU treatment in BRCA2-deficient cells which can be rescued by loss of A3B / UNG / APE1.

- CHIP / Cut & Run should be performed with RELB to show binding of RELB to A3B promoters increased specifically after Cisplatin and HU treatment in BRCA2-deficient cells.

Minor points:

1. In U-ssDNA assays, the staining appears to be pan nuclear and not many foci are visible which can be erroneous. It

would be helpful to measure nuclear intensities and report those instead of the numbers.

2. Gel plots showing the purification of uracil sensor would be helpful.

3. For Figure 1A, it could be helpful to have a full length UNG2-domain structure above and the Delta-UNG (shown) below it to make it more clear.

4. 53BP1 nuclear bodies that form around lesions generated by replication stress during the preceding cell cycle don't usually manifest as pan nuclear 53BP1 staining. They are observed as clear 53BP1 nuclear bodies or foci. The authors could address why they see pan stained nuclei. Additional Cyclin A staining should help characterize whether these are G1-phase cells or not.

5. In Fig. S3h, better SMUG1 blot should be used.

Reviewer #3

(Remarks to the Author)

Duan and colleagues investigate the potential for single stranded DNA induced by BRCA2-deficiency to enable APOBEC3-mediated DNA damage in cancer cells. This manuscript relies heavily on fluorescent detection of uracil or abasic sites in ssDNA using either a catalytically inactive UNG or aldehyde reactive probe respectively, to quantify the amount of the lesions in BRCA2-deficient cells after treatment with replication blocking drugs. The authors found increased uracil and abasic site signal in BRCA2-deficient cells treated with cisplatin or hydroxyurea (HU). The signal was repeated in multiple different cell lines. Uracil signal was dependent on BRCA2-deficiency, APOBEC3B proficiency, and drug treatment. APOBEC3A or APOBEC3B over-expression could restore uracil signal in APOBEC3B KO cells. Abasic sites were increased in BRCA2-deficient cells treated with drug and required UNG2 activity. BRCA2-deficient cells treated with cisplatin or HU also increased DNA double strand breaks (depend on APEX1) and micronuclei formation. Based on these results, the authors propose that genetic instability caused by BRCA2-deficiency may be mediated through an increase of replication stress induced ssDNA that induces APOBEC3B and provides a substrate for cytosine deamination. The authors provide some correlative data to support this in BRCA2-deficient ovarian cancer patients having worse survival if they have high APEX1 expression. While the manuscript contains an exhaustive amount of data to experimentally define the effects they are seeing, I have the following comments that should be addressed before publication.

1) Quantifications of uracil in Figs 1 and 2 are displayed in either foci per nucleus or mean intensity. Please remain consistent and use one or the other.

2) What is being counted as a foci in Figs. 1, 2, 6, and 8 is unclear. The quantifications indicate that the average number of foci per nuclei is around 10 or lower. However, the representative images for uracil foci in BRCA2-deficient cells treated with cisplatin or HU seem to indicate signal throughout the entire nucleus. The images need to be re-evaluated to ensure they are representative of the quantifications. Please also check that each image has similar exposure/background gain. If the background intensity is dependent on the cell conditions, it makes it difficult to assess for accuracy.

3) The major impacts of BRCA2-deficiency on APOBEC-induced uracil seem to require induced replication stress. The authors should discuss whether this is likely to occur in tumors natively or if chemotherapeutic treatment is required.

4) The indication that deamination of BRCA2-induced ssDNA could underly tumor phenotypes is correlative based upon the ovarian cancer analyses. The authors should also evaluate BRCA1-deficient ovarian tumors to determine if the decreased survival is BRCA2 specific. Additionally, is the effect specific to ovarian cancer or is it also seen in breast cancers? The cell line analysis would seem to suggest it should be conserved among cancer types.

5) The proposed model would seem to suggest that BRCA2-deficient cancers would have higher amount of APOBEC-induced mutation. This has been reported in breast cancers but appears to be specific to SBS13 (i.e. abasic site bypass) and is conserved between BRCA1 and BRCA2 deficiency (PMID: 37532520). This should be discussed in the manuscript with a potential explanation.

6) The dependence of uracil induction on APOBEC3B and NFkappaB signaling was only conducted in U2OS cells that have very high APOBEC3B expression and low APOBEC3A expression. Do other cells that have higher endogenous APOBEC3A expression so similar effects dependent on APOBEC3A? If so, then the signaling pathways leading to this response may be different.

7) The manuscript should be checked fully for writing errors. Lines 184-194 are particularly problematic.

Version 1:

Reviewer comments:

Reviewer #1

(Remarks to the Author)

In the revised manuscript most concerns previously raised have been adequately addressed. Listed are a few points that author should consider before finalizing the manuscript:

Related to #2: A better experimental scheme for addressing the cell cycle could have been labeling cells with EdU during Cisplatin/HU treatment and then perusing the wash off followed by IF experiments.

Related to #7: Cyclin A labels both S and G2 phase cells.

Related to #9: This reviewer feels that it is important to include the negative data in Pe01 and CAPAN-1 and discuss the result. This will be quite beneficial to readers while relating the findings in disease context.

Related to #2 in Minor comments: Although the contents of the suggested manuscripts have been included, the references are not cited.

Please check the antibody table: The catalog numbers do not match with the right protein. For eg. Bethyl (A300-246A) is listed for UNG but it is actually RPA32 Ser33
NBP1-49985 is an antibody for UNG and not BRCA2.

Reviewer #2

(Remarks to the Author)

I thank the authors for addressing the major concerns of the paper. Most figure panels have been updated significantly. The authors can proofread the paper once as there are a few typos . E.g: Sup Fig 7 legend title, etc.

Reviewer #3

(Remarks to the Author)

The authors have addressed all my previous comments.

Thank you for the opportunity to respond to Reviewers' comments. We were encouraged by the overall positive feedback and are grateful to all the reviewers for their thoughtful and constructive suggestions. We have carefully addressed each of the comments and have performed extensive new experiments that significantly strengthen our study. In doing so, we uncovered additional findings that not only support our central hypothesis but also offer new mechanistic insights. We are pleased to present these new data and a point-by-point response detailing how each comment has been addressed. Apart from adding the new data to the main manuscript (and referencing it accordingly), additional data that were not included in the revised manuscript are presented at the end of this document and are referred to as "Response Figures".

Major changes/additions are highlighted in blue in the revised manuscript.

REVIEWER 1

Major Comments:

1. A key conceptual gap in this paper is the underlying source of single-strand DNA that distinguishes the phenotype observing between BRCA1- and BRCA2- mutant cells. Functions of BRCA1 and 2 are overall overlapping in the context of reversed fork or gap protection. In contrast, in the context of DSB repair, BRCA1 promotes resection and both BRCA1/BRCA2 contribute to Rad51 loading. Given that authors are observing increased accumulation of DSBs (Fig.5a-c), one likely explanation can be that resected DNA at collapsed forks can account for the phenotype in BRCA2-deficient cells. In contrast, lack of resection in BRCA1-deficient cells suppresses the phenotype. One approach could be testing a BRCA1 mutant (studied extensively by Neil Johnson's lab) proficient in resection but deficient in Rad51 loading.

Authors' comment: We appreciate this suggestion and agree that understanding what drives differences in ssDNA accumulation in BRCA1 vs. BRCA2-deficient settings is important. As suggested by the Reviewer, we obtained *BRCA1* mutant cell lines from Neil Johnson's lab, including the BRCA1 coiled-coil mutant (L1407P-CC), which is proficient in CtIP-mediated resection but deficient in RAD51 loading¹. Unfortunately, these BRCA1 mutant lines are quite sensitive to HU or cisplatin and slow growing when exposed to these agents, which made it difficult to study them in our replication stress-based assays. To address the reviewer's hypothesis from an alternative angle, we depleted CtIP using siRNA in BRCA2-deficient cells to determine whether loss of CtIP-mediated resection would impact U-ssDNA accumulation. If i) CtIP-mediated resection drives ssDNA accumulation in BRCA2-deficient cells and ii)

loss of this function in BRCA1-deficient cells is the reason why BRCA1-deficient cells do not accumulate U-ssDNA, then its loss would be expected to reduce U-ssDNA signal in BRCA2-deficient cells.

- As shown in **Reviewer_Response_Fig. 1**, CtIP depletion did not reduce U-ssDNA accumulation in BRCA2-deficient cells after replication stress. This is consistent with previous work from Sartori group², showing that CtIP does not resect collapsed forks but instead plays a role in fork protection. In line with this, work from Nussenzweig's group has shown that BRCA1 does not participate in resection at single ended or double ended stalled/collapsed forks and instead works downstream of resection³. This is unlike its role in DSB (non-replication associated) resection.
 - While BRCA1 and BRCA2 are both essential for homologous recombination repair of DSBs, increasing evidence suggests their roles at stalled forks differ. For e.g., Vindigni's group has shown that although MUS81 cleaves stalled fork substrate in BRCA2-deficient cells, BRCA1-deficient cells either don't accumulate this stalled-fork associated substrate or are not susceptible to MUS81 dependent cleavage⁴. Similarly, CtIP-mediated stalled fork protection has been shown to synergize with BRCA1 but not BRCA2², further underscoring the differences in BRCA1 vs BRCA2-deficient cells in replication stress contexts. It is not yet clear what drives these differences in BRCA1- vs BRCA2-deficient cells.
 - We agree that fully dissecting the mechanistic differences between BRCA1- and BRCA2-deficient cells, especially with regard to resection and APOBEC activation, is an important future direction. While our current data begin to address this question, a full mechanistic dissection of differences between BRCA1 and BRCA2 lies beyond the scope of this manuscript which focuses on APOBEC3B driven genomic instability in BRCA2-deficient cells. In response to Reviewer's question, we now discuss these differences between BRCA1 and BRCA2 and emphasize the need to investigate the mechanistic differences in the Discussion section (**lines 518-529**).
2. It is imperative that authors demonstrate that Uss-DNA is enriched in S-phase. Differences being observed can be a simple manifestation of alterations in cell cycle upon Cisplatin or HU treatment.

Authors' comment: We agree with the Reviewer that it is important to confirm that the phenotype we observe is S-phase specific. In response, we performed additional analyses and now present data that directly addresses whether uracil accumulation occurs in S-phase. Specifically, we co-stained U-ssDNA-stained cells with PCNA, a well-established marker of active and stressed/stalled replication forks⁵⁻⁷. We found that majority (>80%) of cells with detectable U-ssDNA signal after HU treatment were also PCNA-positive (**Fig. 2a-b**), and around 20-25% of these cells also displayed strong co-localization of U-ssDNA and PCNA, with an average of 10-15 colocalized foci/cell (**Fig. 2c**). These findings strongly support the conclusion that uracil accumulation occurs primarily in replicating (S-phase) cells and at stalled forks. This co-positivity of the PCNA and U-ssDNA signal was more readily detected in HU-treated cells compared to cisplatin-treated cells, likely because cisplatin-treated cells tend to exit S-phase during recovery, albeit with uracil in DNA.

These new results are also consistent with our data showing a strong correlation between RPA and U-ssDNA accumulation in HU and cisplatin treated cells (**Supplementary Fig. 2a-**

c). Together, the findings support that U-ssDNA accumulation upon replication stress occurs specifically in replicating cells. This new data is now presented in the revised manuscript (**Fig. 2a-c; lines 159-170**).

3. AP DNA quantification: The signal needs to be normalized with the input DNA

Authors' comment: The AP DNA signal was indeed normalized to total input DNA using methylene blue staining, which reflects DNA loading. We have now revised the Figure legend to explicitly indicate how input normalization was performed (**Figure legend for Fig. 3**).

4. Establish the specificity of APEX1 siRNAs by either complementing back cDNA or citing its validity from previous studies.

Authors' comment: In this study we used two independent APE1 siRNAs (siAPE1#1 and siAPE1#2). APE1 siRNA #1 has been validated and published previously⁸.

The second APE1 siRNA#2 was ordered as an On-Target siRNA from Dharmacon. As suggested by the Reviewer, to validate the specificity of this siRNA (APE1 siRNA#2), we have now carried out a complementation experiment with APE1 expressing cDNA which was made resistant to APE1 siRNA#2. A western blot showing expression of siRNA#2 resistant APE1 is shown in **Supplementary Fig. 5j**. As shown in **Supplementary Fig. 5k**, re-expression of APE1 in APE1 siRNA#2 treated cells, restored the sensitivity of BRCA2-deficient cells to HU and cisplatin, confirming the specificity of the APE1 siRNA#2. These results have been added to the revised manuscript (**Supplementary Fig. 5j-k**) and the construction of APE1 siRNA resistant APE1 cDNA plasmid is described in Materials and Methods (**lines 1036-1044**).

5. S1 nuclease experiment: S1 nuclease digests both single-strand DNA and RNA. Therefore, utilizing this nuclease to conclude that Uracil is enriched on ss-DNA is misleading. Perhaps complementing the experiment with RNaseA digestion which specifically degrades RNA at C or U residue will be more meaningful.

Authors' comment: We thank the Reviewer for raising this point. While we acknowledge that S1 nuclease digests both ssDNA and RNA, we would like to clarify that the Δ UNG probe we use, just like UNG protein itself, detects uracil specifically in DNA⁹. Therefore, any signal detected with the Δ UNG probe reflects uracil in DNA. Given that UNG does not recognize or act on RNA, Δ UNG probe offers a direct and DNA-specific readout of uracil content.

In order to further confirm this specificity of Δ UNG probe, as suggested by the Reviewer, we also carried out U-ssDNA assay in cells pre-treated with RNaseA. Pre-treatment with RNaseA did not affect the U-ssDNA signal in BRCA2-deficient cells undergoing replication stress. This new data is presented in **Supplementary Fig. 2f-h** and discussed in **lines 183-187**.

6. Fork degradation experiment: The scheme used for this experiment is not logical. Authors use IdU followed by HU and then CIdU. They measure the IdU track as a marker of fork degradation. This scheme assumes that all IdU tracks started at the same time and neglects the

fact that origins don't fire synchronously. Hence the difference being observed can be a mere manifestation of IdU track starting at different time points. A more established way to address fork degradation is IdU, followed by CldU and then HU. Then measure the length of CldU tracks attached to IdU. This way one can ensure that all CldU tracks start at the same time. It is imperative that a head-to-head comparison is performed with BRCA1-mutant cells. Based on established literature one would expect similar results in BRCA1-settings. This would further help parse out if the source of ss-DNA is simply forking collapse or resected fork.

Authors' comment: We appreciate this thoughtful suggestion. The original labeling scheme we used (IdU → HU → CldU) has been employed in previous studies^{10,11} to study both fork degradation and fork restart. Differences in replication origin firing are expected to be equivalent across -HU and +HU conditions. Therefore, any measurable difference in IdU track length after HU treatment should essentially reflect fork degradation. We had used this labeling scheme originally to also detect the effect on fork restart.

Having said that, we do agree that a better approach to exclusively study fork degradation will be to do IdU followed by CldU and then HU. As suggested by the Reviewer, we have now repeated all our DNA fiber experiments using this new scheme. The resulting data is now included in **Supplementary Fig. 4b and 6l**. In addition, to address the Reviewer's point regarding BRCA1-deficient cells, we performed parallel analysis in BRCA1-deficient cells. As shown in **Supplementary Fig. 6l**, while BRCA1-deficient cells also show robust fork degradation, this phenotype is not rescued upon loss of APOBEC3B. These findings further support the specificity of the APOBEC3B effect in the BRCA2-deficient setting.

7. Use of 53BP1 (and that too without any cell cycle marker) as a read out of fork collapse is truly misleading. As authors would agree that 53BP1 marks various lesions and hence using it as a marker of replication fork collapse is incorrect. This reviewer suggests substituting 53BP1 experiment with fork degradation experiments as discussed above.

Authors' comment: We understand the concern and appreciate the feedback. As suggested by the Reviewer, we have revised the fork degradation experiments as described above, using the more appropriate labeling scheme.

In parallel, we have also repeated the 53BP1 assay and now co-stain with Cyclin A to mark S-phase cells. While 53BP1 can mark diverse types of DNA lesions, 53BP1 foci that arise in S-phase cells after replication stress are distinct and well characterized in previous studies and are different than the 53BP1 nuclear bodies formed in G1 cells^{12,13}. The data presented for 53BP1 positive cells in our study is for cells with S-phase specific 53BP1 foci pattern. To confirm that we have now redone the siLuc vs siBRCA2 (HU and cisplatin) experiment to exclusively quantify 53BP1 foci in Cyclin A positive cells to ensure association of 53BP1 pattern we quantify with replication-associated DNA damage. The new 53BP1/cyclin A data is presented in **Supplementary Fig. 4d-f**. In addition, we have revised the manuscript text to refer to 53BP1 as a marker of DNA damage resulting from replication stress, rather than as a direct marker of fork collapse (**lines 265-272**).

8. Fig.5m and Supp 5l. Were all BRCA1 and 2 patient-mutation analyzed established to be pathogenic? If not, one limitation of this analysis can be that many mutations were benign and hence no correlation was observed in the context of BRCA1. It's important that the authors note this limitation in the manuscript.

Authors' comment: Yes, all BRCA1 and BRCA2 mutations included in our analysis are established germline pathogenic variants. To improve clarity, we have now provided a complete list of patient samples and corresponding mutations. As shown in **Supplemental Data 1**, all cases labeled as BRCA1 or BRCA2 mutant, carry confirmed pathogenic driver mutations in these genes.

9. Fig.6k. The finding that A3A/A3B dKO can rescue Cisplatin sensitivity of BRCA2-mutant cells is very striking and unexpected. Typically, A3A and A3B are negligibly expressed in most cell lines and hence assuming that these cytidine deaminases actually contribute to the majority of cisplatin sensitivity is surprising. U2OS cells are known to express A3A and A3B. It is imperative that authors validate these findings in relevant BRCA2-mutant lines as PeO1 and CAPAN-1 to avoid any misleading conclusions.

Authors' comment:

We agree with the Reviewer that it is important to validate the role of A3B in additional BRCA2-deficient lines. To address this, we depleted A3B in both PEO1 and CAPAN1 cells. Additionally, we also tested two non-tumor human mammary epithelial cell lines (HMECs) CP29 and AR20, in which BRCA2 was depleted by siRNA. As shown in **Reviewer Response Fig. 2**, A3B depletion significantly rescued sensitivity of BRCA2-deficient HMECs (CP29 and AR20) to HU and cisplatin, supporting a role for A3B in mediating replication stress-induced cytotoxicity in these settings. However, we see a trend but not a significant rescue in PEO1 and CAPAN-1 cells upon siRNA mediated A3B depletion.

We suggest that this lack of strong rescue in the tumor-derived lines may reflect one or more of the following factors: i) chronic APOBEC3 activity in long established BRCA2 tumor lines like PEO1 and CAPAN1 may have led to compensatory changes or selection of downstream effectors that can help these cells bypass APOBEC3 induced DNA damage; and/or ii) the inherently high baseline genomic instability in these BRCA2-mutant tumor lines may render them intrinsically hypersensitive to HU and cisplatin, to a degree that cannot be effectively rescued by siRNA-mediated A3B depletion alone.

Despite this, the consistent and significant rescue of drug sensitivity in three independent BRCA2-depleted cell lines upon A3B knockdown supports the conclusion that A3B activity contributes to the increased sensitivity to replication stress inducing agents in BRCA2-deficient cells.

10. Why is there an enrichment of U-DNA foci in siLuc, HU+ NFkB samples. Also, to make the conclusion that NFkB reduces U-DNA foci, the correct statistical comparison should be between HU treated vs HU+ NFkB treated siBrca2 samples.

Authors' comment: We agree with the Reviewer that the appropriate statistical comparison is between HU treated vs HU+ NFκBi treated siBRCA2 samples. We have now updated the figure to reflect this comparison and its significance accordingly.

Regarding the enrichment of U-ssDNA foci observed in siLuc HU+NFκBi-treated cells, we were also intrigued by this finding. Our analysis suggests that this increase could be due to NFκB inhibition affecting BRCA2 expression. As shown in the accompanying western blot (Fig. 9j), NFκBi treatment significantly downregulates BRCA2 expression, even in control siLuc cells. This observation is consistent with previous reports¹⁴ showing that NFκB signaling drives BRCA2 transcription. Thus, NFκBi may induce partial BRCA2 loss even in siLuc cells, accounting for the modest increase in U-ssDNA foci seen under HU+NFκBi conditions, thus mimicking siBRCA2 cells treated with HU+NFκBi.

Minor Points:

1. Please include the schematic of treatment and the cell line names in the figures for the ease of readability.

Authors' comment: We have now included these for each of the relevant figures in the main and supplementary figures.

2. Missing Citations: Studies showing direct evidence that APE1 can function at the replication fork and introduce breaks: PMID: 39605722 and 39068174

Authors' comment: Thank you for these relevant suggestions. We have now added these references to the manuscript (**lines 494-496**).

REVIEWER 2

Major Comments:

1. Given that UNG is the primary glycosylase to excise uracils from DNA, it is surprising to see strong uracil signals from the U-ssDNA assay even in the presence of UNG in Figures 1 and 2. The authors also show that cells have increased AP sites in the presence of UNG which reduce after UNG KO in Fig. 3. The authors should ideally show appropriate controls for this assay by adding UNG KO panel. UNG KO experiments have been performed for other assays in subsequent figures but not for the critical U-ssDNA assay.

Authors' comment: We thank the Reviewer for this suggestion. In response, we have now performed the U-ssDNA assay in BRCA2-deficient UNG2 KO cells and uncovered new mechanistic insights that are now part of the revised manuscript.

The straightforward expectation is that UNG2 loss will increase uracil retention in DNA and thus elevate the U-DNA signal. This was indeed true in cells treated with 5FdUR, which incorporates uracil into DNA independently of APOBEC. As expected, UNG2KO cells treated with 5FdUR showed higher U-ssDNA signal than wild-type (**Reviewer_Response_Fig. 3**).

However, interestingly, in BRCA2-deficient UNG2KO cells, we observed reduced U-ssDNA accumulation upon replication stress compared to BRCA2-deficient UNG2 wt cells (**Fig. 8i-j**). While initially unexpected, this result is consistent with our proposed model (**Fig. 10**). In our model, uracil excision by UNG2 and the subsequent processing of the abasic sites at the stalled forks leads to fork collapse and release of cytoplasmic ssDNA fragments which activate NF- κ B signaling and upregulate A3B expression. This upregulated A3B is the major driver of uracil accumulation in BRCA2-deficient cells. Thus, UNG2 is not only involved in uracil removal but is also critical for generating the very ssDNA substrates that trigger A3B upregulation and amplify U-ssDNA accumulation. Thus, in the absence of UNG2, this self-reinforcing positive feedback loop is interrupted upstream of A3B induction, leading to reduced overall U-ssDNA levels.

We carried out additional experiments to test this model and observed:

1. Cytoplasmic ssDNA accumulation is reduced upon loss of UNG2 or APE1 in BRCA2-deficient cells (**Fig. 8a-f**).
2. UNG2 depletion (UNG2 KO and siUNG2) and APE1 depletion (siAPE1) suppresses A3B induction in BRCA2-deficient cells upon replication stress (**Fig. 8g-h**).

Together, these findings underscore that in BRCA2-deficient cells under replication stress, UNG2 is essential not only for uracil processing but also for feeding the positive feedback loop required for APOBEC3B upregulation. We believe this new data adds significant mechanistic depth to our model and strongly supports the central role of the UNG2–APE1–A3B axis in BRCA2-deficient cells undergoing replication stress. This important work is part of the revised manuscript and presented in **Fig. 8** and discussed in Results section (**lines 403-421**).

2. As the authors point out, treatments with cisplatin and HU would cause fork reversal. During fork reversal, nascent DNA strands anneal together limiting the amount of exposed ssDNA. As a result, majority of the cytosines would be end up in dsDNA which cannot be deaminated by A3B. Even if the deamination of cytosines to uracils happened when they were in ssDNA and then forks were reversed, upon reversal, the uracils would be in dsDNA and should not be detectable by U-ssDNA assay.

Authors' comment: As noted by the Reviewer, fork reversal does generate double-stranded DNA structures, which can reduce the amount of exposed ssDNA and potentially limit APOBEC3B-mediated cytosine deamination and/or detection of uracil by our U-ssDNA assay. However, in BRCA2-deficient cells, long stretches of ssDNA at stalled forks is generated by two processes – i) reversed forks which undergo fork degradation^{15,16}, as well as ii) fork uncoupling. Our model proposes that both degraded reversed forks as well as uncoupled replication forks (which have not undergone fork reversal) contribute to the uracil-containing ssDNA signal detected in BRCA2-deficient cells.

In keeping with our model, work from Massimo Lopes's group¹⁵ shows that loss of BRCA2 results in two-fold reduction in reversed forks and a simultaneous increase in ssDNA accumulation at stalled forks. We propose that this ssDNA along with that generated upon fork uncoupling is the substrate for A3B mediated cytosine deamination.

3. The ssDNA in BRCA2-deficient cells after cisplatin or HU would be quickly protected by RPA. How does RPA bound ssDNA get deaminated by APOBEC3B?

Authors' comment: This is an important mechanistic point. While RPA rapidly coats ssDNA, work published by Chelico group has shown that both A3A and A3B can displace RPA from ssDNA to access cytosines and catalyze deamination¹⁷. Although A3A and A3B can displace RPA, A3B's specific activity is reduced approximately 2-fold in the presence of RPA, whereas A3A activity is largely unaffected¹⁷. These findings suggest that APOBEC enzymes retain the capacity to engage with RPA-coated ssDNA, particularly under conditions where ssDNA persists or accumulates. Additional work from Roberts group has shown that although A3A can displace RPA and deaminate the ssDNA, RPA binding is more inhibitory to A3A activity on linear ssDNA than on hairpin structures, highlighting the importance of DNA conformation in modulating deamination efficiency¹⁸. A recent comprehensive review¹⁹ discusses this competitive interaction between RPA coated ssDNA and APOBECs in greater detail. We have now addressed this in the Discussion section of the revised manuscript (**lines 490-491**).

4. Fork degradation in BRCA2-deficient cells is via MRE11/EXO1-mediated exonuclease activity. In Supplementary figure 4, why does loss of UNG or APE1 i.e the increased presence of uracils or AP sites affect exonuclease activity of MRE11/EXO1? Similarly, In Supplementary Figure 6i, the fork degradation phenotype is completely reversed by A3B loss. These results from the fork degradation assays are a bit puzzling. As a subpoint to that, the labeling scheme needs to be modified. Based on the information in the legend, there should be a horizontal black line showing HU treatment of 3hrs in between IdU and CldU. Moreover, if only IdU tracts are being measured, what is the reason to add CldU? Are the authors measuring fork degradation, stalled forks, fork restart?

Authors' comment: We thank the Reviewer for the opportunity to clarify our model. We do not propose that loss of UNG2 or APE1 directly impacts the nuclease activity of MRE11 or EXO1. Rather we propose a sequential process in which fork resection by nucleases such as MRE11 and EXO1 at stalled forks generate stretches of ssDNA. This is in addition to ssDNA generated upon fork uncoupling. Such ssDNA at stalled forks serves as the substrate for APOBEC3B-mediated cytosine deamination. Subsequent processing by UNG2 and APE1 results in DNA breaks further exacerbating fork collapse. Thus, we propose that APOBEC/UNG/APE1 proteins act downstream of MRE11/EXO1-mediated degradation and amplify fork collapse.

Regarding the fiber assay design – we have now modified the assay (c.f Q #6 of Reviewer 1) and as suggested by the Reviewer, we have added a schematic to more clearly represent the experimental workflow and labeling timeline (**Supplementary Fig. 4b and 6l**). Our plan was to study fork restart along with fork degradation, hence the original labeling scheme.

5. Increased replication stress in BRCA2-deficient cells after cisplatin and HU treatment caused increased A3B expression. The authors make a claim that it is due to increased export of DNA in the cytoplasm leading to the activation of NF κ B pathway. However, no direct experiment provides evidence for this claim. Transfected ssDNA leads to A3B activation (Fig. 7h-m) and RELB nuclear translocation (Fig. S7C) but that is completely different from nuclear DNA being released into the cytoplasm after replication stress causing NF κ B activation and increased A3B expression. To solidify these conclusions, the authors need to perform several key experiments.
- i) The authors should show IF images showing A3B and RELB activation in the same nuclei after cisplatin or HU treatment in BRCA2-deficient cells.
 - ii) PLA assay with EdU (nascent nuclear DNA) and actin (cytosolic marker) to show increased cytosolic DNA after cisplatin or HU treatment in BRCA2-deficient cells which can be rescued by loss of A3B / UNG / APE1.
 - iii) CHIP / Cut & Run should be performed with RELB to show binding of RELB to A3B promoters increased specifically after Cisplatin and HU treatment in BRCA2-deficient cells.

Authors' comment: We agree with the Reviewer that providing direct mechanistic evidence of endogenous ssDNA accumulating in the cytoplasm will help support our proposed model. Specifically, our hypothesis that ssDNA fragments generated upon fork collapse are exported to the cytoplasm, where it activates the NF- κ B pathway and subsequently induces A3B expression. In response to the Reviewer's suggestions, we have now conducted several key experiments to rigorously evaluate this pathway:

- i) *Co-presence of RELB and A3B in nuclei after replication stress:*
As requested by the Reviewer, we tested whether A3B and RELB are co-expressed in the same nuclei after cisplatin and HU treatment in BRCA2-deficient cells. Such co-presence would lend support to a model where RELB contributes to A3B upregulation. We have now carried out this experiment and present this data in **Fig. 9d-h**, where we show that co-presence of A3B and RELB in the nuclei of BRCA2-deficient cells is strongly induced upon replication stress.
- ii) *Detection of endogenous cytosolic ssDNA upon replication stress and its dependence on A3B, UNG2 and APE1.*
As suggested by the Reviewer, we carried out PLA assay to study the accumulation of endogenous ssDNA in the cytoplasm of BRCA2-deficient cells upon replication stress. We performed the PLA assay using a validated ssDNA antibody^{20,21} together with actin as a cytoplasmic marker. As shown in **Fig. 8a-c**, we observe a strong induction of PLA signal in the cytoplasm specifically in BRCA2-deficient cells subjected to replication stress (cisplatin or HU treatment), whereas control cells do not show such an increase. These results provide direct support for our model which proposes accumulation of ssDNA in the cytoplasm of BRCA2-deficient cells under replication stress conditions. Importantly, co-depletion of A3B, UNG2, or APE1 significantly reduced accumulation of cytoplasmic ssDNA in these cells (**Fig. 8d-f**), suggesting that accumulation of cytoplasmic ssDNA in BRCA2-deficient cells is dependent on the activity of A3B,

UNG2 and APE1. These new findings directly support our model and are now part of our revised manuscript. We appreciate the reviewer's suggestion, which helped us significantly strengthen the experimental evidence for this aspect of our model.

iii) *RELB binding to the APOBEC3B promoter:*

We agree that demonstrating direct RELB occupancy at the A3B promoter would further strengthen our proposed mechanism. We did try the CHIP experiment with the RELB antibody we currently have but were not able to get this antibody to work for CHIP in our cells. Given the substantial time and resource commitment required for testing additional RELB antibodies and APOBEC promoter specific primers, we have not included this experiment in the current study but agree that it would be valuable to pursue in future work specifically aimed at further dissecting the RELB driven APOBEC3B upregulation in BRCA2-deficient cells. We would like to note that (i) prior literature has already established RELB as a transcriptional regulator of APOBEC3B^{22,23}, and (ii) our data now demonstrate strong co-expression and nuclear co-localization of RELB and A3B under replication stress conditions. These findings provide indirect but compelling support for RELB-mediated A3B transcription. Should the Reviewer feel that this experiment is essential for the current study, we are open to revisiting this experiment to test additional RELB antibodies.

Minor Points:

1. In U-ssDNA assays, the staining appears to be pan nuclear and not many foci are visible which can be erroneous. It would be helpful to measure nuclear intensities and report those instead of the numbers.

Authors' comment: We would like to note that pan-nuclear staining was observed primarily in BRCA2-mutant tumor lines PEO1 and CAPAN1. For all other cases, U-ssDNA staining predominantly appeared as discrete nuclear foci.

We compared U-ssDNA quantification by both foci number and mean intensity in cell lines where both metrics could be reliably measured. As shown in **Supplementary Fig. 1d-e**, the results from intensity-based quantification closely match the foci-based analysis, supporting the robustness of our findings across different readout methods.

2. Gel plots showing the purification of uracil sensor would be helpful.

Authors' comment: We now provide the uracil sensor purification gel (**Supplementary Fig. 1a**).

3. For Figure 1A, it could be helpful to have a full length UNG2-domain structure above and the Delta-UNG (shown) below it to make it more clear.

Authors' comment: We now provide a schematic of full length UNG2 above the mutant Delta-UNG version (**Fig. 1a**).

4. 53BP1 nuclear bodies that form around lesions generated by replication stress during the preceding cell cycle don't usually manifest as pan nuclear 53BP1 staining. They are observed as clear 53BP1 nuclear bodies or foci. The authors could address why they see pan stained nuclei. Additional Cyclin A staining should help characterize whether these are G1-phase cells or not.

Authors' comment: The 53BP1 nuclear bodies are, as the Reviewer mentioned, a result of DNA damage in the previous cell cycle^{12,13}. These are not the lesions we are analyzing. The 53BP1 foci we are analyzing represent DNA damage associated with collapsed forks in S phase, rather than nuclear bodies formed from lesions carried over from the preceding cell cycle. Such replication stress associated 53BP1 foci have been studied previously^{12,13}. As suggested by the Reviewer, to confirm the cell cycle phase of the cells being analyzed for 53BP1 foci formation, we performed co-staining with Cyclin A, a well-established S-phase marker. As shown in **Supplementary Fig. 4e**, the 53BP1-positive cells we are quantifying are also cyclin A-positive.

Please also refer to our response to Reviewer #1 Question 7.

5. In Fig. S3h, better SMUG1 blot should be used.

Authors' comment: We have now provided a better SMUG1 western blot (**Supplementary Fig. 3h**).

REVIEWER 3

Major Comments:

1. Quantifications of uracil in Figs 1 and 2 are displayed in either foci per nucleus or mean intensity. Please remain consistent and use one or the other.

Authors' comment: We thank the Reviewer for this suggestion. In our study, we have quantified U-ssDNA by counting foci per nucleus. However, in BRCA2-mutant tumor lines PEO1 and CAPAN1, the U-ssDNA staining appeared more pan-nuclear, making it difficult to reliably quantify discrete foci. For this reason, only for B2-mutant tumor lines we used mean nuclear intensity as the readout.

As mentioned in response to Reviewer 2, Minor Point Q1 - To ensure consistency and validate our approach, we compared U-ssDNA quantification by both foci number and mean intensity in cell lines where both metrics could be reliably measured. As shown in **Supplementary Fig. 1d-e**, the results from intensity-based quantification closely match the foci-based analysis, supporting the robustness of our findings across different readout methods. These comparisons

confirm that the observed increases in U-ssDNA accumulation in BRCA2-deficient cells are consistent regardless of whether foci number or mean intensity is used. We have now clarified this distinction in the revised manuscript (lines 144-145).

2. What is being counted as a foci in Figs. 1, 2, 6, and 8 is unclear. The quantifications indicate that the average number of foci per nuclei is around 10 or lower. However, the representative images for uracil foci in BRCA2-deficient cells treated with cisplatin or HU seem to indicate signal throughout the entire nucleus. The images need to be re-evaluated to ensure they are representative of the quantifications. Please also check that each image has similar exposure/background gain. If the background intensity is dependent on the cell conditions, it makes it difficult to assess for accuracy.

Authors' comment: To clarify our methodology, we have now included a more detailed description of how U-ssDNA foci were quantified in the Materials and Methods section of the revised manuscript. We would also like to reassure the Reviewer that all images within each experimental set were taken using identical exposure settings to avoid variability due to image capture parameters. This ensures that any differences in signal intensity in our samples reflect biological differences rather than technical artifacts.

Image quantification details: Briefly, to quantify the number of U-ssDNA foci per nucleus, we used ImageJ's automated "Find Maxima" function, which identifies discrete signal peaks based on user-defined parameters such as "Prominence" (i.e., the minimum intensity difference from surrounding pixels required to be counted as a peak/foci). In our original analysis, the Prominence threshold was set at 40 to apply a more stringent filter and reduce the risk of counting background noise, resulting in an average of approximately 10–15 foci per cell in BRCA2-deficient cells undergoing replication stress. This setting likely underestimates the number of visible foci in the representative images.

To test the robustness of our analysis, we repeated the quantification for all the different datasets using a lower prominence setting of 30, which increased the number of foci detected per nucleus. Importantly, this change did not alter the overall statistical conclusions of our study: BRCA2-deficient cells treated with cisplatin or HU still showed a significantly higher number of foci compared to untreated controls. This consistency was observed across all experimental conditions and figure panels, supporting the reliability of our findings regardless of the exact Prominence threshold used. We have now re-analyzed all datasets with this lower Prominence setting of 30 and present that data in our manuscript.

Link of ImageJ "Find Maxima" information: <https://imagej.net/ij/docs/guide/146-29.html>

3. The major impacts of BRCA2-deficiency on APOBEC-induced uracil seem to require induced replication stress. The authors should discuss whether this is likely to occur in tumors natively or if chemotherapeutic treatment is required.

Authors' comment: We thank the reviewer for this thoughtful comment. We agree that our current study relies on induced replication stress (via cisplatin and HU) to reveal the

relationship between BRCA2 deficiency and APOBEC-induced genomic instability. However, BRCA2-deficient cells show elevated levels of replication stress due to defects in fork protection and high genome instability. This endogenous replication stress likely provides sufficient ssDNA substrate to induce and engage APOBEC enzymes in BRCA2-deficient tumors. Chemotherapeutic treatments, we believe, further amplify this effect. Thus, the APOBEC-driven uracil accumulation and genomic instability we describe is likely to occur in both untreated and treated BRCA2-deficient tumors. We have added this point to the Discussion section of the revised manuscript (**lines 580-587**).

4. The indication that deamination of BRCA2-induced ssDNA could underly tumor phenotypes is correlative based upon the ovarian cancer analyses. The authors should also evaluate BRCA1-deficient ovarian tumors to determine if the decreased survival is BRCA2 specific. Additionally, is the effect specific to ovarian cancer or is it also seen in breast cancers? The cell line analysis would seem to suggest it should be conserved among cancer types.

Authors' comment: Thank you for bringing up this point. We did analyze the BRCA1-deficient ovarian tumors and that data is presented in **Supplementary Fig. 5n**. As seen here, unlike patients with BRCA2-deficient tumors, BRCA1-deficient cases do not show any correlation with APE1 levels. Additionally, as suggested by the Reviewer, we also looked into Breast Cancer datasets. We worked with our collaborator Dr. Birkbak to interrogate *BRCA2* mutant breast cancer dataset (METABRIC). We find that although the trend is the same (low APE1 levels = poor survivorship), it does not reach significance like it does in the ovarian cancer dataset. We believe that this could be because unlike in ovarian cancer, *BRCA2* mutation carriers with breast cancer do not get cisplatin/carboplatin or even PARPi as first line of treatment²⁴. This could confound the analysis for us because although low APE1 levels contribute to increased resistance to cisplatin induced killing (as shown in our study), alteration in APE1 levels is not expected to affect hormonal therapy – which will be the first line of treatment for *BRCA2* mutant breast cancer cases which most often are ER+.

We have now added this to our Discussion (**lines 572-579**).

5. The proposed model would seem to suggest that BRCA2-deficient cancers would have higher amount of APOBEC-induced mutation. This has been reported in breast cancers but appears to be specific to SBS13 (i.e. abasic site bypass) and is conserved between BRCA1 and BRCA2 deficiency (PMID: 37532520). This should be discussed in the manuscript with a potential explanation.

Authors' comment: We propose that *BRCA1/2* mutant tumors are still more closely linked to the HRD signature and/or mutational signature 3^{24,25} than to the APOBEC signature. Given that APOBEC signature is a result of mis-reading of U-DNA by the DNA polymerase resulting mostly in C>T mutations (SBS2 signature) and mis-reading of abasic sites resulting in C>G/C>A mutations (SBS13 signature)²⁶⁻²⁸, we reason that there might not necessarily be an enrichment for the APOBEC signature in BRCA2-deficient tumors. As shown in our study, APOBEC action on ssDNA results in double strand breaks. In the context of defective homologous recombination repair in BRCA2-deficient cells, these DSBs are more likely to

result in genomic deletions, insertions, and rearrangements—hallmarks of the HRD signature, rather than a high burden of point mutations that define the APOBEC signature.

This mechanistic pathway may help reconcile the observations from Mertz et al., which show enrichment of SBS13 in both BRCA1- and BRCA2-deficient tumors²⁹.

We propose that:

- In BRCA2-deficient tumors, APOBEC activity may drive mutagenesis through DSB formation and indels, not predominantly through base substitution.
- In BRCA1-deficient tumors, which may preserve fork protection to some degree, there might be sufficient ssDNA exposure and repair competency to support bypass of abasic lesions, leading to SBS13 accumulation.
- Interestingly, there is no enrichment (instead there is reduction) in SBS2 APOBEC signature in BRCA1 deficient cells while there seems to be a significant enrichment of this signature in BRCA2-deficient tumors. This would be in line with our model which suggests an increase in APOBEC activity in BRCA2-deficient cells compared to BRCA1-deficient cells.

Thus, despite differences in APOBEC induction between BRCA1 and BRCA2 contexts, the shared enrichment of SBS13 across both may reflect common downstream lesion processing, rather than equivalent APOBEC activation.

6. The dependence of uracil induction on APOBEC3B and NFkappaB signaling was only conducted in U2OS cells that have very high APOBEC3B expression and low APOBEC3A expression. Do other cells that have higher endogenous APOBEC3A expression show similar effects dependent on APOBEC3A? If so, then the signaling pathways leading to this response may be different.

Authors' comment: It is certainly of interest whether APOBEC3A plays a similar role in driving genomic instability in BRCA2-deficient cells. At this point, we have focused mostly on APOBEC3B driven events – especially given published reports that have indicated a role of APOBEC3B early in tumorigenesis³⁰⁻³². It is possible that other cells that have higher endogenous APOBEC3A might show similar phenotype. We have briefly addressed this by working with BICR6 cell line which is known to express A3A. As show in **Reviewer_Response_Fig. 4**, we do find that in these cells, accumulation of U-ssDNA is driven by A3A. It is possible that in cells expressing both A3A and A3B, there is some redundancy with respect to their roles, especially in context of targeting ssDNA at stalled forks. It has been shown previously that A3A expression is also NF-κB dependent³³⁻³⁵.

7. The manuscript should be checked fully for writing errors. Lines 184-194 are particularly problematic.

Authors' comment: We apologize for this oversight and have ensured that the manuscript is fully checked for any potential writing errors.

Response Figures:

[REDACTED]

Reviewer Response Figure 2. Loss of A3A/A3B rescues cell sensitivity in BRCA2-deficient human mammary epithelial cells (HMECs). (a) Western blot analysis of BRCA2 and A3B in whole cell lysates of MEC lines, AR20 and CP29, transfected with the indicated siRNAs for 48hrs. Vinculin serves as the loading control. (b) CellTiter-Glo based cell survival assay to determine the sensitivity of HMEC lines, AR20 and CP29, transfected with the indicated siRNAs for 48hrs prior to treatment with cisplatin (left) or HU (right). Error bars represent SD between triplicates.

Reviewer Response Figure 3. 5-FdUR treatment leads to higher U-ssDNA accumulation in UNG2 KO compared to UNG2 wild type cells. (a) Representative images of U-ssDNA accumulation in U2OS wildtype (WT) or U2OS UNG2 KO cells treated with 20 μ M of 5-FdUR for 48hrs. (b) Quantification of U-ssDNA mean intensity by immunostaining with 3xFLAG- Δ UNG probe under denaturing conditions. WT or UNG2 KO cells were treated as described in (a). SuperPlots of 2 independent experiments (n=2) are represented per condition as indicated. Number of U-ssDNA foci per cell was analyzed through Image J. >100 cells were analyzed in each replicate. Each highlighted shape (●▲) represents the average of each replicate with the black lines representing the mean \pm SD of n=2 independent experiments. Statistical significance was determined by using repeated measurement model followed by multiple comparisons with Bonferroni Post hoc test. ***p \leq 0.001.

[REDACTED]

REFERENCES

1. Nacson, J., Kraiss, J.J., Bernhardt, A.J., Clausen, E., Feng, W., Wang, Y., Nicolas, E., Cai, K.Q., Tricarico, R., Hua, X., et al. (2018). BRCA1 Mutation-Specific Responses to 53BP1 Loss-Induced Homologous Recombination and PARP Inhibitor Resistance. *Cell Rep* 24, 3513-3527 e3517. 10.1016/j.celrep.2018.08.086.
2. Przetocka, S., Porro, A., Bolck, H.A., Walker, C., Lezaja, A., Trenner, A., von Aesch, C., Himmels, S.F., D'Andrea, A.D., Ceccaldi, R., et al. (2018). CtIP-Mediated Fork Protection Synergizes with BRCA1 to Suppress Genomic Instability upon DNA Replication Stress. *Mol Cell* 72, 568-582 e566. 10.1016/j.molcel.2018.09.014.
3. Pavani, R., Tripathi, V., Vrtis, K.B., Zong, D., Chari, R., Callen, E., Pankajam, A.V., Zhen, G., Matos-Rodrigues, G., Yang, J., et al. (2024). Structure and repair of replication-coupled DNA breaks. *Science* 385, eado3867. 10.1126/science.ado3867.
4. Lemacon, D., Jackson, J., Quinet, A., Brickner, J.R., Li, S., Yazinski, S., You, Z., Ira, G., Zou, L., Mosammaparast, N., and Vindigni, A. (2017). MRE11 and EXO1 nucleases degrade reversed forks and elicit MUS81-dependent fork rescue in BRCA2-deficient cells. *Nat Commun* 8, 860. 10.1038/s41467-017-01180-5.
5. Ciccica, A., Nimonkar, A.V., Hu, Y., Hajdu, I., Achar, Y.J., Izhar, L., Petit, S.A., Adamson, B., Yoon, J.C., Kowalczykowski, S.C., et al. (2012). Polyubiquitinated PCNA recruits the ZRANB3 translocase to maintain genomic integrity after replication stress. *Mol Cell* 47, 396-409. 10.1016/j.molcel.2012.05.024.
6. Mirsanaye, A.S., Typas, D., and Mailand, N. (2021). Ubiquitylation at Stressed Replication Forks: Mechanisms and Functions. *Trends Cell Biol* 31, 584-597. 10.1016/j.tcb.2021.01.008.
7. Vujanovic, M., Krietsch, J., Raso, M.C., Terraneo, N., Zellweger, R., Schmid, J.A., Tagliatalata, A., Huang, J.W., Holland, C.L., Zwicky, K., et al. (2017). Replication Fork Slowing and Reversal upon DNA Damage Require PCNA Polyubiquitination and ZRANB3 DNA Translocase Activity. *Mol Cell* 67, 882-890 e885. 10.1016/j.molcel.2017.08.010.
8. Pines, A., Bivi, N., Vascotto, C., Romanello, M., D'Ambrosio, C., Scaloni, A., Damante, G., Morisi, R., Filetti, S., Ferretti, E., et al. (2006). Nucleotide receptors stimulation by extracellular ATP controls Hsp90 expression through APE1/Ref-1 in thyroid cancer cells: a novel tumorigenic pathway. *J Cell Physiol* 209, 44-55. 10.1002/jcp.20704.
9. Palinkas, H.L., Bekesi, A., Rona, G., Pongor, L., Papp, G., Tihanyi, G., Holub, E., Poti, A., Gemma, C., Ali, S., et al. (2020). Genome-wide alterations of uracil distribution patterns in human DNA upon chemotherapeutic treatments. *Elife* 9. 10.7554/eLife.60498.
10. Schlacher, K., Christ, N., Siaud, N., Egashira, A., Wu, H., and Jasin, M. (2011). Double-strand break repair-independent role for BRCA2 in blocking stalled replication fork degradation by MRE11. *Cell* 145, 529-542. 10.1016/j.cell.2011.03.041.
11. Leuzzi, G., Marabitti, V., Pichierri, P., and Franchitto, A. (2016). WRNIP1 protects stalled forks from degradation and promotes fork restart after replication stress. *EMBO J* 35, 1437-1451. 10.15252/embj.201593265.
12. Lukas, C., Savic, V., Bekker-Jensen, S., Doil, C., Neumann, B., Pedersen, R.S., Grofte, M., Chan, K.L., Hickson, I.D., Bartek, J., and Lukas, J. (2011). 53BP1 nuclear bodies form around DNA lesions generated by mitotic transmission of chromosomes under replication stress. *Nat Cell Biol* 13, 243-253. 10.1038/ncb2201.

13. Fielden, J., Siegner, S.M., Gallagher, D.N., Schroder, M.S., Dello Stritto, M.R., Lam, S., Kobel, L., Schlapansky, M.F., Jackson, S.P., Cejka, P., et al. (2025). Comprehensive interrogation of synthetic lethality in the DNA damage response. *Nature* *640*, 1093-1102. 10.1038/s41586-025-08815-4.
14. Wu, K., Jiang, S.W., Thangaraju, M., Wu, G., and Couch, F.J. (2000). Induction of the BRCA2 promoter by nuclear factor-kappa B. *J Biol Chem* *275*, 35548-35556. 10.1074/jbc.M004390200.
15. Mijic, S., Zellweger, R., Chappidi, N., Berti, M., Jacobs, K., Mutreja, K., Ursich, S., Ray Chaudhuri, A., Nussenzweig, A., Janscak, P., and Lopes, M. (2017). Replication fork reversal triggers fork degradation in BRCA2-defective cells. *Nat Commun* *8*, 859. 10.1038/s41467-017-01164-5.
16. Tagliatalata, A., Alvarez, S., Leuzzi, G., Sannino, V., Ranjha, L., Huang, J.W., Madubata, C., Anand, R., Levy, B., Rabadan, R., et al. (2017). Restoration of Replication Fork Stability in BRCA1- and BRCA2-Deficient Cells by Inactivation of SNF2-Family Fork Remodelers. *Mol Cell* *68*, 414-430 e418. 10.1016/j.molcel.2017.09.036.
17. Adolph, M.B., Love, R.P., Feng, Y., and Chelico, L. (2017). Enzyme cycling contributes to efficient induction of genome mutagenesis by the cytidine deaminase APOBEC3B. *Nucleic Acids Res* *45*, 11925-11940. 10.1093/nar/gkx832.
18. Brown, A.L., Collins, C.D., Thompson, S., Coxon, M., Mertz, T.M., and Roberts, S.A. (2021). Single-stranded DNA binding proteins influence APOBEC3A substrate preference. *Sci Rep* *11*, 21008. 10.1038/s41598-021-00435-y.
19. Wong, L., Sami, A., and Chelico, L. (2022). Competition for DNA binding between the genome protector replication protein A and the genome modifying APOBEC3 single-stranded DNA deaminases. *Nucleic Acids Res* *50*, 12039-12057. 10.1093/nar/gkac1121.
20. Li, S., Kong, L., Meng, Y., Cheng, C., Lemacon, D.S., Yang, Z., Tan, K., Cheruiyot, A., Lu, Z., and You, Z. (2023). Cytosolic DNA sensing by cGAS/STING promotes TRPV2-mediated Ca(2+) release to protect stressed replication forks. *Mol Cell* *83*, 556-573 e557. 10.1016/j.molcel.2022.12.034.
21. Feng, S., Marhon, S.A., Sokolowski, D.J., D'Costa, A., Soares, F., Mehdipour, P., Ishak, C., Loo Yau, H., Ettayebi, I., Patel, P.S., et al. (2024). Inhibiting EZH2 targets atypical teratoid rhabdoid tumor by triggering viral mimicry via both RNA and DNA sensing pathways. *Nat Commun* *15*, 9321. 10.1038/s41467-024-53515-8.
22. Periyasamy, M., Singh, A.K., Gemma, C., Farzan, R., Allsopp, R.C., Shaw, J.A., Charmsaz, S., Young, L.S., Cunnea, P., Coombes, R.C., et al. (2021). Induction of APOBEC3B expression by chemotherapy drugs is mediated by DNA-PK-directed activation of NF-kappaB. *Oncogene* *40*, 1077-1090. 10.1038/s41388-020-01583-7.
23. Leonard, B., McCann, J.L., Starrett, G.J., Kosyakovsky, L., Luengas, E.M., Molan, A.M., Burns, M.B., McDougale, R.M., Parker, P.J., Brown, W.L., and Harris, R.S. (2015). The PKC/NF-kappaB signaling pathway induces APOBEC3B expression in multiple human cancers. *Cancer Res* *75*, 4538-4547. 10.1158/0008-5472.CAN-15-2171-T.
24. Shah, J.B., Pueschl, D., Wubbenhorst, B., Fan, M., Pluta, J., D'Andrea, K., Hubert, A.P., Shilan, J.S., Zhou, W., Kraya, A.A., et al. (2022). Analysis of matched primary and recurrent BRCA1/2 mutation-associated tumors identifies recurrence-specific drivers. *Nat Commun* *13*, 6728. 10.1038/s41467-022-34523-y.
25. Polak, P., Kim, J., Braunstein, L.Z., Karlic, R., Haradhavala, N.J., Tiao, G., Rosebrock, D., Livitz, D., Kubler, K., Mouw, K.W., et al. (2017). A mutational signature reveals alterations

- underlying deficient homologous recombination repair in breast cancer. *Nat Genet* 49, 1476-1486. 10.1038/ng.3934.
26. Nik-Zainal, S., Davies, H., Staaf, J., Ramakrishna, M., Glodzik, D., Zou, X., Martincorena, I., Alexandrov, L.B., Martin, S., Wedge, D.C., et al. (2016). Landscape of somatic mutations in 560 breast cancer whole-genome sequences. *Nature* 534, 47-54. 10.1038/nature17676.
 27. Wang, Y., Robinson, P.S., Coorens, T.H.H., Moore, L., Lee-Six, H., Noorani, A., Sanders, M.A., Jung, H., Katainen, R., Heuschkel, R., et al. (2023). APOBEC mutagenesis is a common process in normal human small intestine. *Nat Genet* 55, 246-254. 10.1038/s41588-022-01296-5.
 28. Alexandrov, L.B., Kim, J., Haradhvala, N.J., Huang, M.N., Tian Ng, A.W., Wu, Y., Boot, A., Covington, K.R., Gordenin, D.A., Bergstrom, E.N., et al. (2020). The repertoire of mutational signatures in human cancer. *Nature* 578, 94-101. 10.1038/s41586-020-1943-3.
 29. Mertz, T.M., Rice-Reynolds, E., Nguyen, L., Wood, A., Cordero, C., Bray, N., Harcy, V., Vyas, R.K., Mitchell, D., Lobachev, K., and Roberts, S.A. (2023). Genetic inhibitors of APOBEC3B-induced mutagenesis. *Genome Res* 33, 1568-1581. 10.1101/gr.277430.122.
 30. Caswell, D.R., Gui, P., Mayekar, M.K., Law, E.K., Pich, O., Bailey, C., Boumelha, J., Kerr, D.L., Blakely, C.M., Manabe, T., et al. (2024). The role of APOBEC3B in lung tumor evolution and targeted cancer therapy resistance. *Nat Genet* 56, 60-73. 10.1038/s41588-023-01592-8.
 31. Lee, J.Y., Schizas, M., Geyer, F.C., Selenica, P., Piscuoglio, S., Sakr, R.A., Ng, C.K.Y., Carniello, J.V.S., Towers, R., Giri, D.D., et al. (2019). Lobular Carcinomas In Situ Display Intralesion Genetic Heterogeneity and Clonal Evolution in the Progression to Invasive Lobular Carcinoma. *Clin Cancer Res* 25, 674-686. 10.1158/1078-0432.CCR-18-1103.
 32. Venkatesan, S., Angelova, M., Puttick, C., Zhai, H., Caswell, D.R., Lu, W.T., Dietzen, M., Galanos, P., Evangelou, K., Bellelli, R., et al. (2021). Induction of APOBEC3 Exacerbates DNA Replication Stress and Chromosomal Instability in Early Breast and Lung Cancer Evolution. *Cancer Discov* 11, 2456-2473. 10.1158/2159-8290.CD-20-0725.
 33. Oh, S., Bournique, E., Bowen, D., Jalili, P., Sanchez, A., Ward, I., Dananberg, A., Manjunath, L., Tran, G.P., Semler, B.L., et al. (2021). Genotoxic stress and viral infection induce transient expression of APOBEC3A and pro-inflammatory genes through two distinct pathways. *Nat Commun* 12, 4917. 10.1038/s41467-021-25203-4.
 34. Roelofs, P.A., Martens, J.W.M., Harris, R.S., and Span, P.N. (2023). Clinical Implications of APOBEC3-Mediated Mutagenesis in Breast Cancer. *Clin Cancer Res* 29, 1658-1669. 10.1158/1078-0432.CCR-22-2861.
 35. Isozaki, H., Sakhtemani, R., Abbasi, A., Nikpour, N., Stanzione, M., Oh, S., Langenbucher, A., Monroe, S., Su, W., Cabanos, H.F., et al. (2023). Therapy-induced APOBEC3A drives evolution of persistent cancer cells. *Nature* 620, 393-401. 10.1038/s41586-023-06303-1.

We thank all the reviewers for their evaluation and suggestions that helped strengthen this manuscript.

REVIEWERS' COMMENTS

Reviewer #1 (Remarks to the Author):

In the revised manuscript most concerns previously raised have been adequately addressed. Listed are a few points that author should consider before finalizing the manuscript:

Authors' comment: We thank the reviewer for all the insightful feedback.

1. Related to #2: A better experimental scheme for addressing the cell cycle could have been labeling cells with EdU during Cisplatin/HU treatment and then perusing the wash off followed by IF experiments.

Authors' comment: We appreciate this suggested experiment to address the cell cycle stage. However, we have previously carried out the described experiment but were not able to successfully see the EdU signal in any of the cells irrespective of the U-ssDNA staining. We suspect that given the long-term treatment and recovery time we use for cisplatin/HU treatment (30-48h), any EdU signal that is acquired during the staining stages gets diluted or lost by the time we capture and fix the cells.

The current Cyclin A and U-ssDNA co-staining allows us to successfully address this question in HU treated cells.

2. Related to #7: Cyclin A labels both S and G2 phase cells.

Authors' comment: We thank the reviewer for raising this point and have included this information in the manuscript (lines 271-272).

3. Related to #9: This reviewer feels that it is important to include the negative data in Pe01 and CAPAN-1 and discuss the result. This will be quite beneficial to readers while relating the findings in disease context.

Authors' comment: We acknowledge the importance of addressing the effect of A3A/A3B in BRCA2-mutant tumor cell lines. We have now added colony survival data to Supplementary Figure 6, panel (o-q) for PEO1 upon loss of A3A/A3B.

4. Related to #2 in Minor comments: Although the contents of the suggested manuscripts have been included, the references are not cited.

Authors' comment: We thank the reviewer for the relevant suggested references. However, we did add these references (PMID: 39605722 and 39068174). Please refer to line number 497-498.

5. Please check the antibody table: The catalog numbers do not match with the right protein. For eg. Bethyl (A300-246A) is listed for UNG but it is actually RPA32 Ser33 NBP1-49985 is an antibody for UNG and not BRCA2.

Authors' comment: We apologize for this error and now ensure that all catalog numbers are correctly listed for each antibody.

Reviewer #2 (Remarks to the Author):

I thank the authors for addressing the major concerns of the paper. Most figure panels have been updated significantly. The authors can proofread the paper once as there are a few typos . E.g: Sup Fig 7 legend title, etc.

Authors' comment: We thank the reviewer for all the feedback. We apologize for the typos in the manuscript and have now fixed those errors and have fully checked the manuscript for any other typos/errors.

Reviewer #3 (Remarks to the Author):

The authors have addressed all my previous comments.

Authors' comment: We thank the reviewer for confirming that all their previous comments have been addressed and for their feedback during the Review cycle.